# Mountain gorillas maintain strong affiliative biases for maternal siblings despite high male reproductive skew and extensive exposure to paternal kin

Nicholas M Grebe[1]*, Jean Paul Hirwa[2], Tara S Stoinski[2], Linda Vigilant[3], Stacy Rosenbaum[1]

[1]Department of Anthropology, University of Michigan-Ann Arbor, Ann Arbor, United States; [2]The Dian Fossey Gorilla Fund International, Atlanta, United States; [3]Department of Primatology, Max Planck Institute for Evolutionary Anthropology, Leipzig, Germany

**Abstract** Evolutionary theories predict that sibling relationships will reflect a complex balance of cooperative and competitive dynamics. In most mammals, dispersal and death patterns mean that sibling relationships occur in a relatively narrow window during development and/or only with same-sex individuals. Besides humans, one notable exception is mountain gorillas, in which non-sex-biased dispersal, relatively stable group composition, and the long reproductive tenures of alpha males mean that animals routinely reside with both maternally and paternally related siblings, of the same and opposite sex, throughout their lives. Using nearly 40,000 hr of behavioral data collected over 14 years on 699 sibling and 1235 non-sibling pairs of wild mountain gorillas, we demonstrate that individuals have strong affiliative preferences for full and maternal siblings over paternal siblings or unrelated animals, consistent with an inability to discriminate paternal kin. Intriguingly, however, aggression data imply the opposite. Aggression rates were statistically indistinguishable among all types of dyads except one: in mixed-sex dyads, non-siblings engaged in substantially more aggression than siblings of any type. This pattern suggests mountain gorillas may be capable of distinguishing paternal kin but nonetheless choose not to affiliate with them over non-kin. We observe a preference for maternal kin in a species with a high reproductive skew (i.e. high related-ness certainty), even though low reproductive skew (i.e. low relatedness certainty) is believed to underlie such biases in other non-human primates. Our results call into question reasons for strong maternal kin biases when paternal kin are identifiable, familiar, and similarly likely to be long-term groupmates, and they may also suggest behavioral mismatches at play during a transitional period in mountain gorilla society.

*For correspondence:
ngrebe@umich.edu

**Competing interest:** The authors declare that no competing interests exist.

## Editor's evaluation

This study fundamentally advances our understanding of the ways in which kin recognition might operate in complex social societies using a compelling longitudinal dataset of a wild mountain gorilla population. Relying on detailed behavioural and pedigree data for 157 individuals, and robust statistical analyses, the authors clearly show affiliative biases among maternal siblings with aggression being more likely for unrelated siblings. Given the structure of mountain gorilla society, this research calls into question the assumptions underlying maternal kin preferences, particularly when paternal kin may be distinguishable, and therefore provides a valuable contribution to discussions on mechanisms of kin selection and kin discrimination.

## Introduction

In humans and non-human animals alike, sibling social relationships are marked by continuous dynamics of conflict and cooperation that begin before birth (*Trivers, 1974*), and can persist throughout an entire lifespan, with important fitness consequences for the individuals involved (*Lu, 2007*; *Hudson and Trillmich, 2008*; *Nitsch et al., 2013*). While classical frameworks of sibling interactions emphasized competition among brood- or litter-mates for limited parental resources during times of dependency (e.g. *Mock and Parker, 1997*), subsequent developments across numerous academic disciplines demography: e.g., *Sear and Mace, 2008*; *Nitsch et al., 2013*; sociology: e.g., *Steelman et al., 2002*; *Lu, 2007*; behavioral ecology: e.g., *Silk, 2002*; *Hudson and Trillmich, 2008*; developmental psychology: e.g., *Lamb and Sutton-Smith, 2014* have explored the full arc of sibling competition and cooperation across the lifespan and demonstrated the complexity and diversity inherent to sibling relationships. In understanding the evolution of human sibling relationships in particular, comparative studies of our primate cousins have identified a number of factors that predict when and how siblings cooperate and compete. Inconsistent results within and between species, along with the remarkable flexibility of human social systems; however, limits the translational value of many primate models. Here, we address these gaps by presenting an extensive longitudinal study of wild mountain gorillas (*Gorilla beringei beringei*), an endangered great ape whose unique, flexible social structure serves as a valuable comparative model for humans.

Classic models of kin selection predict that the social/mating structure of animal groups creates patterns of relatedness between group members, which then selects for kin recognition mechanisms that manifest in differences in cooperative, affiliative, competitive, and/or aggressive behavior (*Hamilton, 1964*; *Grafen, 1990*; *Mateo and Hauber, 2015*). This straightforward idea has spawned a large body of work on kin selection that spans animal groups (reviewed in *Widdig, 2007*; *Smith, 2014*). Within primates in particular, investigations of kin discrimination have yielded notably mixed results. Some studies report no bias towards related individuals (e.g. for food-sharing in long-tailed macaques: *Schaub, 1996*; for various affiliative and aggressive behaviors in yellow baboons: *Erhart et al., 1997*). Others support the existence of sophisticated kin discrimination based at least partially on phenotype matching, perhaps in interaction with familiarity, as evidenced by social partner preferences (*Wu et al., 1980*), rates of affiliative behavior (*Smith et al., 2003*; *Streich et al., 2002*; *Smith et al., 2003*), or visual gaze biases (*Pfefferle et al., 2014*). Still others find support for familiarity alone as the determinant of interaction patterns, with familiarity being assessed by time spent living together (in pig-tailed macaques: *Fredrickson and Sackett, 1984*), common clustering around mothers (in ursine colobus: *Wikberg et al., 2014*), or age differences (in bonnet macaques: *Silk, 1994*; in chimpanzees: *Langergraber et al., 2007*). There is also extensive debate over differences in patterns of discrimination between paternally and maternally related kin (e.g. *Chapais, 2001*). Some have suggested that non-monogamous primates evince matrilineal but not patrilineal sibling kin discrimination, perhaps due to restrictive dispersal patterns that may limit opportunities for long-term social relationships with paternal kin (*Mitani et al., 2000*; *Langergraber et al., 2007*) or polygynandrous mating systems with low reproductive skew that complicate efforts to accurately identify them (*Galezo et al., 2022*). Yet other perspectives challenge a clean distinction between maternal and paternal kin, suggesting that complex interactions between familiarity and other modes of kin discrimination structure social bonds across primates (see e.g. *Silk, 2002*; *Streich et al., 2002*; *Widdig et al., 2006*; *Silk, 2009*; *Lynch et al., 2017*).

As one of the main contributors to familiarity, age differences within sibling and non-sibling dyads might influence social dynamics (*Widdig et al., 2001*; *Streich et al., 2002*; *Langergraber et al., 2007*; *Pfefferle et al., 2014*; *Wikberg et al., 2014*). On one hand, siblings close in age might be more likely to compete for limited parental resources (*Tung et al., 2016*; *Salmon and Hehman, 2021*); on the other hand, as longer-lasting co-residents within the same family environment, they might also be expected to form stronger affiliative bonds than siblings distant in age (though, again, this may not apply equally to maternal and paternal sibships; *Streich et al., 2002*). It is unclear to what extent age proximity effects are restricted to genetic relatives. Female rhesus macaques appear to bias affiliation towards similarly aged peers, even when unrelated to them (*Widdig et al., 2001*). Among female baboons, even in individuals not related through the matriline, dyadic bond strength weakened with increasing age differences; however, when analyses are restricted to females unrelated through both the matriline and patriline, effects of age differences attenuated sharply (*Smith et al., 2003*; *Silk*

*et al., 2006*). These results once again imply social familiarity (as indexed by age differences) and kin discrimination are both important for predicting sibling relationship qualities (*Godoy et al., 2016*), though their additive and/or interactive effects remain poorly defined.

Finally, the sex makeup of the dyad might influence interaction styles due to the differential benefits males and females receive from interactions with brothers, sisters, and unrelated partners (e.g. *Lonsdorf et al., 2018*). For males, especially in species who engage in aggressive intrasexual competition, other males, brothers included can represent important allies (e.g. *Meikle and Vessey, 1981*; *Goodall, 1986*) or rivals (*Daly and Wilson, 1988*; *Snowdon and Pickhard, 1999*; *Chagnon et al., 2017*) during status-striving efforts in adulthood. In either case, assessing physical capacities or formidability would aid in these efforts. Accordingly, rough-and-tumble play between males might serve as a rehearsal for intrasexual competition in adulthood (*Gray, 2019*), suggesting such a behavior should occur most often in male-male relationships—a prediction supported by research on male-dominant primates (e.g. *Brown and Dixson, 2000*; *Maestripieri and Ross, 2004*). While male-male interaction patterns might generally differ from those of other sex configurations, these differences may themselves partially depend on kinship. In chimpanzees, some evidence suggests that fraternal relationships among adolescents and adults are more affiliative and cooperative than relationships between unrelated males (e.g. *Mitani, 2009*; *Sandel et al., 2020*). From the female perspective, evidence for fraternal influences on fitness outcomes is mixed. While there are some reports of chimpanzee brothers 'adopting' immature sisters (*Hobaiter et al., 2014*; *Reddy and Mitani, 2019*), and one demographic study of humans reports benefits of older brothers on women's lifetime fitness (*Nitsch et al., 2013*), other research on primate species with sex-biased dispersal suggests no lasting fitness effects (except perhaps when mothers die; *Engh et al., 2009*).

From the perspective of both males and females, sisters may represent important future alloparental helpers, either for the individual themselves (e.g. *Hamilton et al., 1982*; *Gould, 2000*; *Hobaiter et al., 2014*), or the individual's offspring (e.g. *Johnson et al., 1980*; *Nishida, 1983*; but see *Silk et al., 2006*). Thus, cultivating relationships with sisters via affiliative interactions might be beneficial for both males and females. Last, for females in particular, sororal relationships may exert important influences on future rank and resource acquisition outcomes (*Charpentier et al., 2008*; *Lea et al., 2014*; cf. *Engh et al., 2009*). However, these kinds of sex-biased interactions might additionally depend on age differences between siblings (*Lonsdorf et al., 2018*), once again underscoring the complicated mix of demographic factors that may influence sibling relationships.

Understanding the nature and evolution of complex social relationships, such as those between siblings, requires long-term investigations that reveal how individuals respond behaviorally to socio-ecological variation (e.g. *Alberts and Altmann, 2012*). Additionally, comparative models for human sibling relationships in particular are most useful when the model species shares key social features. With these principles in mind, mountain gorillas in particular are a compelling model for studies of sibling dynamics. First, long-term monitoring of wild mountain gorillas by the Dian Fossey Gorilla Fund has revealed social structures marked by extensive diversity in relatedness, age proximity, and sex makeup infrequently observed in other non-human primate groups (*Robbins et al., 2009b*; *Roy et al., 2014*). Mountain gorillas regularly form multi-female, single-male groups, as well as multi-female, multi-male groups in which multiple males reproduce, though paternity data and unsophisticated paternal kin discrimination mechanisms are consistent with historically high reproductive skew (*Bradley et al., 2005*; *Rosenbaum et al., 2015*; *Vigilant et al., 2015*). As a result of their highly variable social structure, researchers regularly observe co-resident immatures that are full siblings, paternal half-siblings, maternal half-siblings, or unrelated to one another.

Second, like humans but unlike nearly all other primates, both male and female mountain gorillas may opt to disperse or remain in their natal groups upon reaching maturity (*Robbins et al., 2009b*; *Stoinski et al., 2009*). This permits fraternal, sororal, and mixed-sex relationships that can last for an entire lifespan, meaning that siblings can be an important source of support well into adulthood. For example, adult males benefit greatly from allies who help them fend off outside male challengers and prevent females from transferring out of their groups (*Sicotte, 1993*; *Rosenbaum et al., 2016b*; *Mirville, 2018*). Adult females with offspring benefit from male protection from infanticide (*Harcourt and Stewart, 2007*; *Robbins et al., 2013*) in multi-male groups, this protection could potentially come not only from mates, but from brothers as well, who receive indirect fitness benefits from their sister's reproductive success. While the benefits of female-female relationships remain understudied in

this species, higher-ranking females have better energy balance and shorter inter-birth intervals, and it is plausible that support from other females plays a role in achieving and maintaining dominance (*Robbins et al., 2005*; *Wright et al., 2014*; *Wright et al., 2020*). At the same time, sibling bonds in mountain gorillas are characterized by the potential for competition as well as cooperation. Even after co-resident mountain gorilla siblings cease competing over parental resources, they are likely to compete over other limited resources, including dominance positions, mating opportunities, and preferred foods (*Harcourt and Stewart, 2007*), which highlights the complexity of making straightforward predictions about affiliative and aggressive interactions and relatedness (e.g. *Silk et al., 2010*).

In the present study, we use nearly 40,000 hr of behavioral data spanning 14 years to describe patterns of interactions between siblings and demographically comparable non-sibling dyads in social groups of wild mountain gorillas. Using extensive maternity and genetic paternity data available for 157 identifiable individuals studied from late infancy through adulthood, we examine whether full siblings, maternal half-siblings (hereafter, 'maternal siblings'), paternal half-siblings ('paternal siblings'), and unrelated co-residents ('non-siblings') exhibit differing patterns of affiliation (playing, grooming, and time spent in close proximity) and agonism (contact and non-contact aggression) in line with models of kin selection, after adjusting for the potential mediating presence of mothers in these interactions. Currently, there are no data regarding kin discrimination patterns among mountain gorilla siblings. While research in other primate species has reported both greater cooperation and greater competition between siblings compared to non-siblings, on balance we expect that mountain gorillas should be more cooperative/affiliative with siblings than non-siblings due to the potential for inclusive fitness benefits. Regarding matrilineal versus patrilineal kin biases, it is unclear whether gorilla siblings should exhibit the same maternal sibling bias observed in other primate species. In species with meaningfully lower reproductive skew, paternal sibling discrimination may be challenging or of limited use due to dispersal patterns that limit opportunities to benefit from paternal kin. Gorillas' high reproductive skew, in contrast, could conceivably facilitate paternal sibling discrimination—but it could also obviate the need to develop any such mechanisms. Given these competing considerations, we ask which scenario is more consistent with the dynamics of gorillas' sibling relationships.

In addition to kinship, we also investigate the impacts of age differences and sex. We determine whether age differences, commonly used in kinship research as a proxy for familiarity between social partners, predict patterns of affiliative or agonistic behavior, and whether these patterns differ between paternal and maternal kin (as some evidence from cercopithecine monkeys suggests; e.g. *Streich et al., 2002*). Finally, we test whether male-male, female-female, and mixed-sex sibling relationships are characterized by differing rates and types of social interactions, and whether these sex category differences are restricted to kin. While developmental changes in social relationships are not the focus of this study, these comparisons do speak to questions regarding the kinds of benefits siblings might be expected to deliver later in life for one's status-striving and/or resource acquisition efforts: e.g., among males, are fraternal relationships marked by higher rates of playing and fighting, and sororal relationships higher rates of grooming?; among females, are sororal relationships marked by the highest rates of grooming compared to any other dyad configuration?; are affiliative patterns unique to siblings, or are comparable trends found in unrelated dyads?

## Results
### Affiliative behaviors

In our full sample of 1934 unique dyads spanning 7832 dyad-years, full siblings (n=43 dyads) and maternal siblings (n=101 dyads) played and groomed each other significantly more than did paternal siblings (n=555 dyads) or non-siblings (n=1235 dyads; all comparisons p<0.05; *Figure 1A and B*). Age differences (in our sample, mean: 5.85 years; SD: 4.53 years; range: 0–23.5 years) interacted with relatedness in predicting grooming (p=0.017), but not play (p=0.427). Play consistently dropped for siblings and non-siblings alike as age differences increased (γ ranging from –0.31 to –0.37, all p<0.001; *Figure 2A*). By contrast, grooming rates were relatively unrelated to age differences between partners (γ ranging from –0.09 to 0.00, all p>0.05; *Figure 2B*).

Male-male dyads (n=503) played more than either mixed-sex (n=977) or female-female dyads (n=454); conversely, female-female dyads groomed each other more than either mixed-sex or male-male dyads (all p<0.001; *Figure 1C and D*). These patterns for play and grooming were strongly

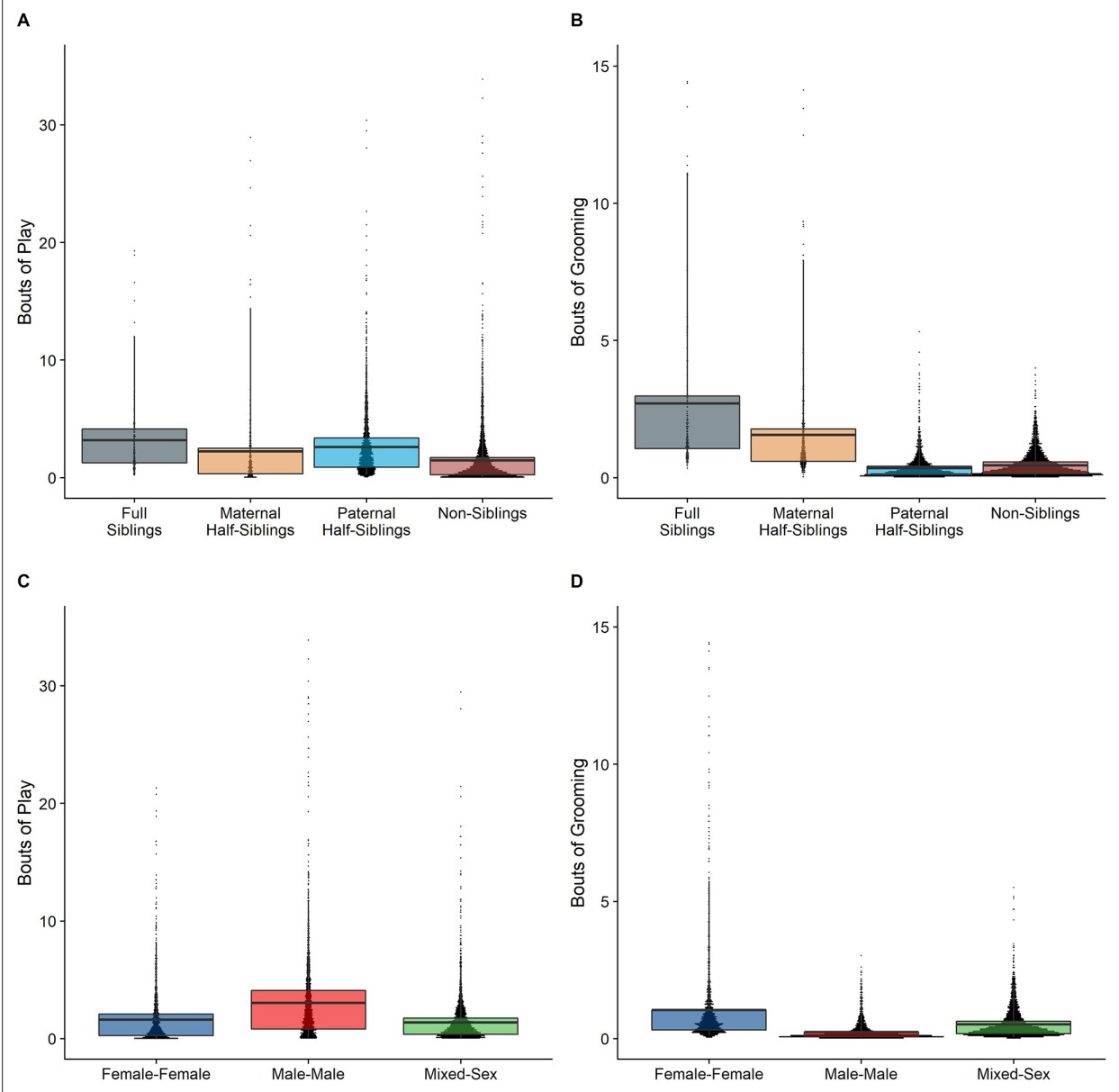

**Figure 1.** Box and dot plots comparing relatedness categories (**A, B**) and sex categories (**C, D**) for play rates (left) and grooming rates (right).

The online version of this article includes the following figure supplement(s) for figure 1:

**Figure supplement 1.** Box and dot plots for play within gorilla dyads that have 'early life familiarity' only (n=6724), separated by relatedness and sex category.

**Figure supplement 2.** Box and dot plots for grooming within gorilla dyads that have 'early life familiarity' only (n=6724), separated by relatedness and sex category.

moderated by age differences (ps<0.006) and were significantly, albeit more weakly, moderated by relatedness (p=0.012 and 0.045, respectively). Play dropped rapidly with increasing age differences ($\gamma$ = −0.37 to −0.30) for all sex configurations (all p<0.001; *Figure 3A*). Grooming was steadily low in male-male and mixed-sex dyads ($\gamma$ = −0.02 and −0.03, p>0.32), though it dropped with increasing age differences in female-female dyads ($\gamma$ = −0.11, p<0.001), such that differences between sex categories became indistinguishable after approximately a 10-year age difference (*Figure 3B*). Play was consistently highest among male-male dyads within all relatedness categories, with the exception of maternal siblings, who exhibited more comparable rates of play among male-male and mixed-sex dyads (*Figure 1—figure supplement 1*). Grooming was also consistently highest among

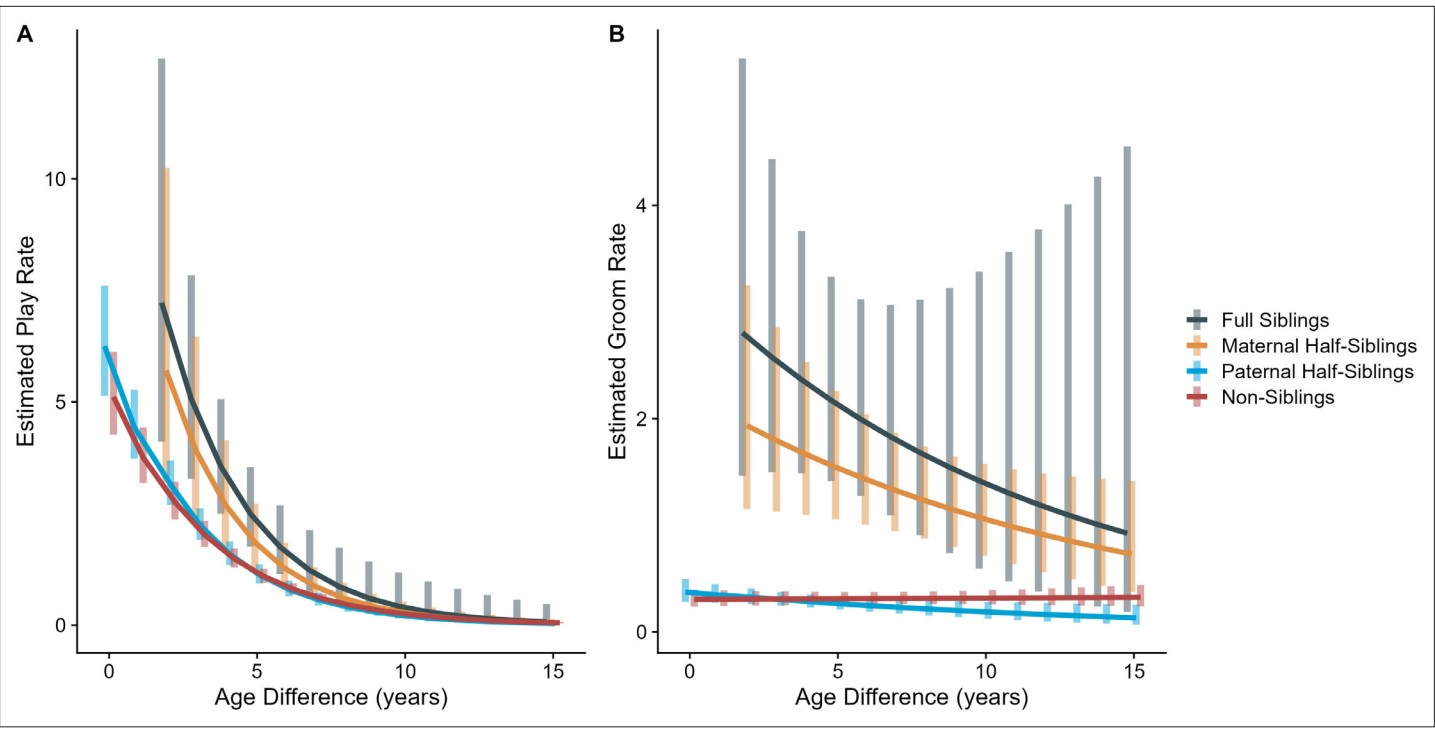

**Figure 2.** Estimated rates of play (**A**) and grooming (**B**) across a range of age differences, separated by relatedness category. Bars represent 95% CI for rates of behavior at a given age difference.

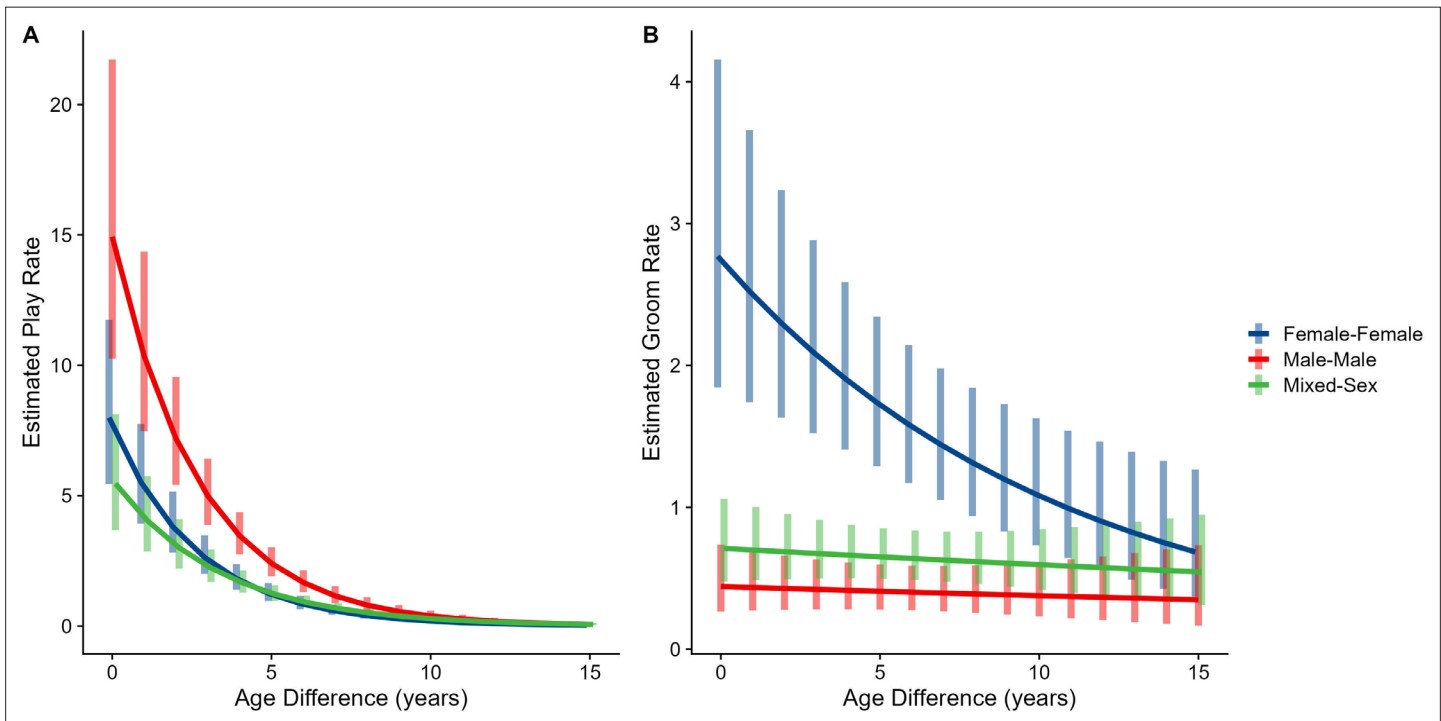

**Figure 3.** Estimated rates of play (**A**) and grooming (**B**) across a range of age differences, separated by sex category. Bars represent 95% CI for rates of behavior at a given age difference.

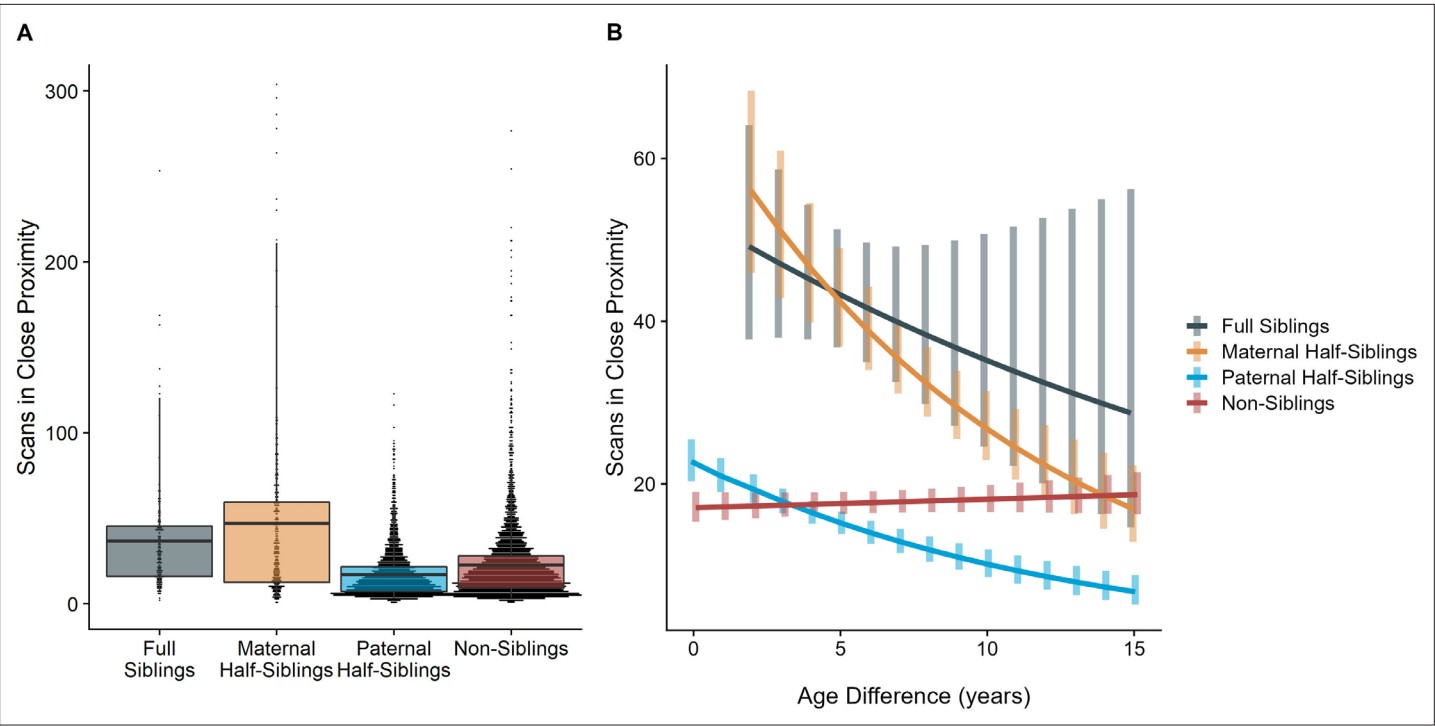

**Figure 4.** Box and dot plots (**A**) and estimated trends across a range of age differences (**B**) for the time gorilla dyads spent in close proximity, separated by relatedness category. Bars in (**B**) represent 95% CI for rates of proximity at a given age difference.

female-female dyads of all types, though the magnitude of this difference varied between relatedness categories (*Figure 1—figure supplement 2*).

When restricting analyses to dyads who lived in the same social group during the first year of the younger partner's life (n=6724 dyad-years), results for affiliative behavior remained qualitatively similar: full and maternal siblings played and groomed more than paternal siblings or non-siblings (see *Appendix 1—tables 1 and 2*). Interactions between relatedness categories and sex makeup weakened and were no longer significant (for play: p=0.387; for grooming: p=0.347); sex category × age difference interactions remained significant for grooming (p<0.001) and for play (p=0.031). Additionally, all reported results were robust to the inclusion of average age of the dyad as a covariate (Appendix 3).

## Time spent in proximity

The time dyads spent in close proximity (<2 m) with each other also varied between relatedness categories (p<0.001), with maternal siblings and full siblings once again spending more time near each other than non-siblings, who themselves spent more time in close proximity than paternal siblings did (all comparisons p<0.001; *Figure 4A*). However, these patterns too were moderated by age differences (p<0.001). Proximity decreased with increasing age differences in maternal siblings and paternal siblings (γ = –0.08 and –0.09, p<0.001), but did not decrease significantly in full siblings or non-siblings (γ = –0.04 and 0.01, p>0.29). Thus, while all classes of siblings spent more time near each other than non-siblings when near in age, even when adjusting for their mother's presence, this distinction was partially reversed at large age differences, when paternal siblings spent much less time near each other than any other dyad category (*Figure 4B*).

When restricting analyses to dyads who lived in the same social group during the first year of the younger partner's life, results for proximity remained consistent: full and maternal siblings spent more time together than paternal siblings or non-siblings, though this difference was reversed at large age differences in the same manner as our primary model (see *Appendix 1—tables 1 and 2* for full results).

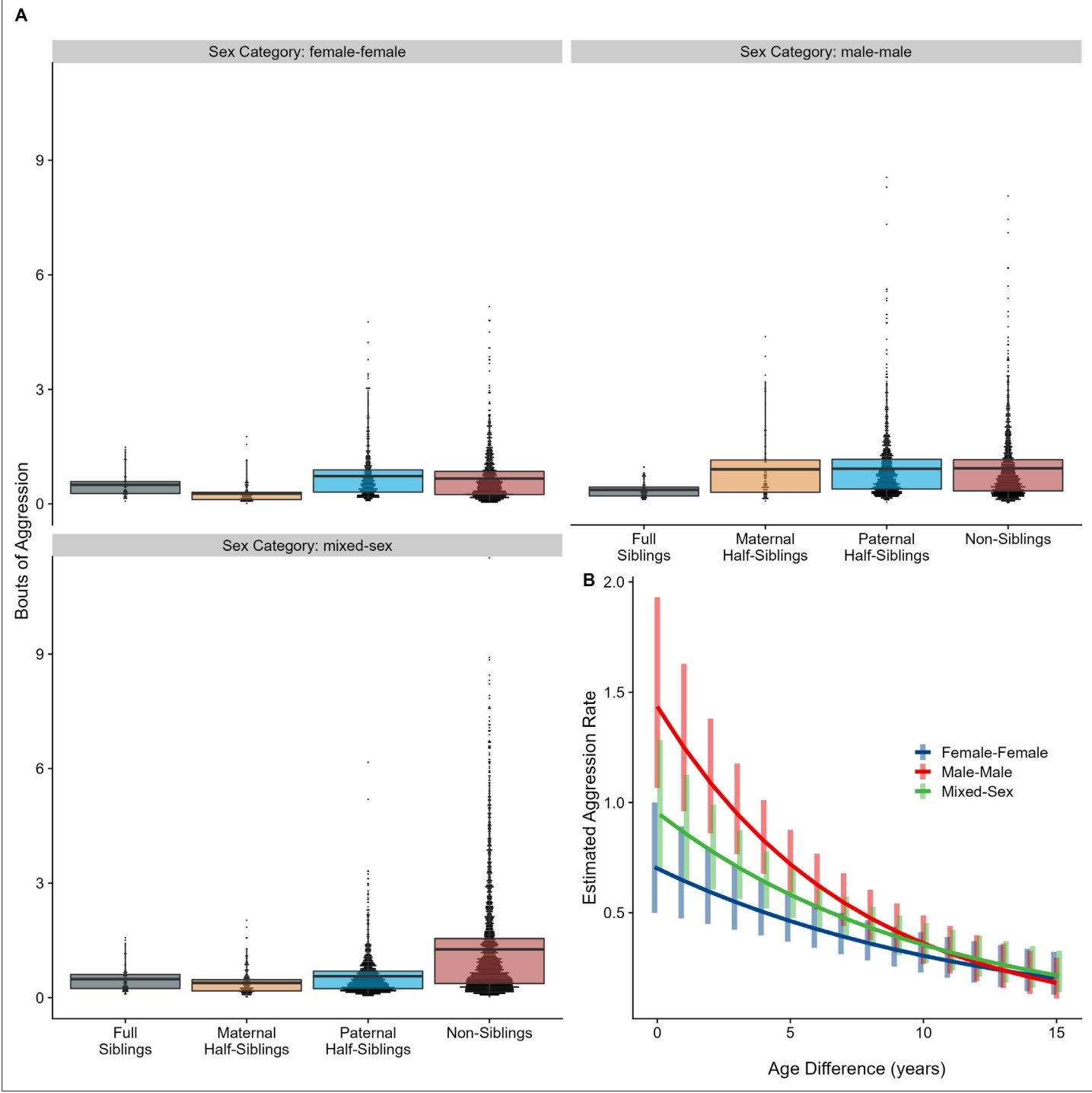

**Figure 5.** Box and dot plots (**A**) and estimated trends across a range of age differences (**B**) for aggression within gorilla dyads, separated by relatedness and sex category. Bars in (**B**) represent 95% CI for rates of aggression at a given age difference.

The online version of this article includes the following figure supplement(s) for figure 5:

**Figure supplement 1.** Box and dot plots for aggression within gorilla dyads that have 'early life familiarity' only (n=6724), separated by relatedness and sex category.

## Competitive behaviors

Neither relatedness nor sex category on their own significantly predicted rates of aggressive behavior (p=0.876 and 0.838, respectively). However, our model did reveal a significant sex makeup × relatedness interaction term (p=0.025; *Figure 5A*). Decomposing this interaction, among female-female and male-male dyads, there were no statistically significant contrasts between relatedness categories. In mixed-sex dyads, non-siblings engaged in substantially more aggression than any sibling category (all p<0.050). While sex category also interacted with age differences in predicting aggression (p=0.009), marginal trends were consistently negative (γ = −0.14 to −0.08, p<0.001), such that, across sex categories, dyads more distant in age engaged in less aggression than dyads closer in age (*Figure 5*). All reported results were robust to the inclusion of average age of the dyad as a covariate (Appendix 3).

Unlike results for affiliative behavior, when restricting analyses to dyads who lived in the same social group during the first year of the younger partner's life, results for aggressive behavior qualitatively shifted. Relatedness and sex categories no longer significantly interacted (p=0.444), and the significant pairwise differences between siblings and non-siblings in mixed-sex dyads shrank markedly and were no longer significant (all p>0.60). Age difference remained a highly significant predictor of aggression (p<0.001). Inspection of our data confirmed that much aggression specifically occurred in the context of females transferring into groups–that is, when females first encountered unrelated, unfamiliar males–accounting for non-significant pairwise differences among male-female dyads with early-life familiarity (*Figure 5—figure supplement 1*). See *Appendix 1—tables 1 and 2* for full results.

## Discussion

In a comprehensive examination of dyadic mountain gorilla social relationships spanning 14 years and nearly 40,000 hr of observation, we find complex patterns of affiliation and competition within gorilla pairs that speak to sex-, age-, and relatedness-specific social biases. In general, siblings affiliated with each other more and spent more time together than non-siblings, even when accounting for the presence of mothers. But within siblings, affiliative patterns further varied: full and maternal siblings were in most cases much more affiliative than paternal siblings, whose behavior more closely resembled that of non-siblings. We consistently observed a trend for male-male dyads to play more, for female-female dyads to groom more, and for mixed-sex dyads to fall intermediate between these groups. Examining competitive behaviors, on the other hand, revealed a narrower sibling bias. Aggression was most common in mixed-sex non-sibling dyads, and larger age differences similarly predicted less aggression across all dyad types.

At the broadest level, our results support the existence of affiliative biases towards kin in mountain gorillas. While past research has been largely equivocal about the extent of kin discrimination that relies on mechanisms beyond familiarity (e.g. *Wikberg et al., 2014*; *Godoy et al., 2016*; *Lynch et al., 2017*)–and indeed, our results do support a role of familiarity (at least as indexed by age difference) in structuring social interactions–our findings are unlikely to be entirely explained by mere exposure for at least three reasons:

First, gorilla social groups are tight-knit and cohesive compared to their close ape relatives (*Goodall, 1986*; *Remis, 1997*; *Morrison et al., 2021b*; *Schaller and Emlen, 1963*; *Doran and McNeilage, 1998*), reflected by remarkable features such as the ability for these groups to buffer the fitness costs of maternal loss (*Morrison et al., 2021a*). Due to this highly cohesive structure, all individuals in a group, related or not, are very likely quite familiar with one another. Even within this tight-knit context, however, ubiquitous 'play groups' of infants and juveniles frequently form around adult males (*Rosenbaum and Silk, 2022*), providing similar-aged animals even more opportunities to interact and become familiar with one another. Second, prosocial biases towards siblings are not fully explained by the greater familiarity that closer-in-age animals likely have (*Widdig et al., 2001*; *Smith et al., 2003*; *Lynch et al., 2017*). We observed clear biases towards kin at all but the largest age differences–and some siblings in our data set were 20 or more years apart in age–even though sibling and non-sibling age-mates in the same social group would typically be expected to possess close familiarity. Finally, while mothers undoubtedly mediate social interactions of offspring, especially for immature individuals, biases towards siblings persist even when adjusting for the frequency of her presence during interaction periods. Jointly, these considerations suggest a sibling bias in mountain gorillas subject to influence, but not determination, by demographic factors, which we interpret as aiding in the

development of sibling relationships that exist across timescales rarely observed in other non-human primates.

The observation that full and maternal siblings groomed, played, and spent more time near each other than paternal siblings or non-siblings, who tended to affiliate at comparable rates, further suggests that mountain gorillas, like several other primate species studied (*Langergraber et al., 2007*; *Silk et al., 2006*; *Lynch et al., 2017*), evince much stronger maternal than paternal kin bias (see also *Rosenbaum et al., 2015*, who found little evidence for paternal kin discrimination among fathers and offspring). Interestingly, this 'asymmetric bias' in affiliation seems to persist even though the high reproductive skew of mountain gorillas opposes the dynamic thought to underlie such biases in other primate species (i.e. low reproductive skew; *Galezo et al., 2022*). Thus, one question concerns why mountain gorillas do not appear to more strongly favor paternal siblings. Current evidence indicates that single-male gorilla groups across research sites are genetically polygynous (reviewed in *Rosenbaum and Silk, 2022*), and while there can be considerable temporal variation, reproductive skew is generally much higher in multi-male gorilla groups than in, for example, chimpanzee, savannah baboon, or rhesus macaque groups (*Vigilant et al., 2015*; *Surbeck et al., 2017*; *Alberts et al., 2003*; *Widdig et al., 2004*). We propose that, despite possessing a mating system quite unlike these other primate species, mountain gorillas still exhibit a comparable maternal sibling bias due to a mismatch between their historical mating structure–which we infer consisted of highly polygynous one-male units–and their contemporary social structure of tight-knit, often multi-male groups. In other words, while individuals in highly polygynandrous groups might find it too difficult to detect and adjust affiliation toward paternal kin, perhaps mountain gorillas fail to do so because, until very recently, it was unnecessary. If co-residency was enough to identify paternal kin with reasonable accuracy, a more sophisticated recognition mechanism would be unlikely to evolve.

One important limitation of our analyses is that we have far less information about sibling relationships in single-male groups—where shared paternity is nearly or entirely certain—than we do about siblings in multi-male groups, where reproductive skew is high but not 100%. The structure of the research groups during our 14 years of data was primarily multi-male, with only 60 dyad-years of data (n=10 total dyads) from single-male groups. All of these dyads were paternal half-siblings (i.e. there were no unrelated or maternally related dyads available in single-male groups). This small sample size and lack of variability in relatedness precludes a test of whether natal group structure predicts features of sibling relationships. If all available partners are paternal kin, then discrimination becomes a question of full versus paternal siblings, rather than discrimination among four categories (full, half maternal, half paternal, and unrelated). Choice is constrained, perhaps limiting the utility of a discrimination mechanism. It would be interesting to know whether animals in single-male groups treat paternal versus full siblings differently, but the currently available data do not allow us to answer this question.

Notably, while we see little evidence that mountain gorillas show a prosocial bias towards paternal siblings, patterns of aggressive behavior suggest there may still be kin recognition mechanisms at play for all sibling types. Aggression remained low across most combinations of relatedness and sex configurations, with one exception: mixed-sex interactions among non-siblings. This pattern is consistent with males deploying aggression in the context of mate attraction or coercion. Our interpretation is further bolstered by the distinct statistical patterns that emerge for aggression when examining unrelated mixed-sex dyads with early-life familiarity, which suggest much of the mixed-sex aggression in our data took place among unrelated and unfamiliar male-female dyads. These encounters were most common when females transferred into new groups and were thus likely to be particularly attractive mates for males. Past research in gorillas suggests male aggression towards females may have a number of non-mutually exclusive functions: to police female-female aggression, to discourage female dispersal or mate choice, or to indicate protective ability or overall condition (*Robbins, 2009a*; *Breuer et al., 2016*). The fact that this kind of aggression was observed less frequently among related male-female pairs is another observation consistent with accurate kin discrimination. It also suggests active inbreeding avoidance, to the extent that aggression truly serves a mate attraction function. While death and dispersal have been suggested to obviate the need for sophisticated inbreeding avoidance mechanisms in some primates (e.g. baboons; *Galezo et al., 2022*), such an explanation is unlikely to apply to contemporary mountain gorillas. Living with opposite-sex relatives after sexual maturity is a routine occurrence in this species. Prior research confirms strong inbreeding avoidance

between father-daughter dyads in this species (*Vigilant et al., 2015*), but further work is needed to investigate the extent to which male mate choice is manifested via female-directed aggression, and whether females, for their part, possess additional mechanisms to avoid mating with kin, including paternal siblings.

Together, these observations–prosocial biases towards kin that do not appear to be fully explained by familiarity; a stronger maternal than paternal sibling prosocial bias; and avoidance of intersexual aggression across all sibling types–both speak to key questions about the development of great ape sibling relationships and present two additional puzzles for interpretation. First, traditional mechanistic explanations for sibling biases that typically appeal to exposure during developmental periods appear largely inconsistent with our results and the nature of mountain gorilla sociality, in which siblings and non-siblings, and maternal and paternal siblings, are all likely to have significant exposure to one another during development. It is possible that maternally mediated early-life exposure effects that we could not measure–e.g., via repeatedly sharing night nests (*Fossey, 1979*) –function analogously to the manner in which co-residence duration serves as a key component of kin recognition in humans (*Lieberman et al., 2007*), or that preferential mother-father relationships post-birth might lead to social preferences among siblings (*Rosenbaum et al., 2016a*). Individuals may also possess some degree of phenotype matching ability (*Widdig, 2007*; *Parr et al., 2010*; *Langergraber, 2012*; *Pfefferle et al., 2014*).

Second, the lack of evidence for a prosocial bias towards paternal siblings is not readily reconciled with clear behavioral evidence of reduced aggression within these same dyads. This remarkable disjunct between apparent sibling recognition and sibling bias suggests that from a mountain gorilla's perspective, paternal siblings are known entities that nevertheless are less attractive social partners than maternal siblings, despite each being equal relatives. There may be multiple, non-mutually exclusive explanations for this dynamic. Perhaps the presence of paternal siblings provides fewer benefits to an individual than do other sibling types–this possibility, while previously suggested (e.g. *Cords et al., 2018*), has not been systematically investigated and is an ideal target for future research. Relatedly, the strength of maternal versus paternal kin bias may to some degree depend on the costs of the behaviors in question (*Widdig et al., 2006*). A mismatch between historical and current social structure might also lead to inconsistent, weakened kin recognition among paternal siblings that manifests in the contrasting patterns we report. Ultimately, disentangling these potential explanations within a species that only exists in the wild may depend on the opportunity to study long-term mating patterns and the impacts of 'natural experiments' such as early maternal loss or adoption (most often carried out by adult males in this species; *Fossey, 1979*; *Morrison et al., 2021a*).

## Conclusion

Our analyses of sibling relationships in mountain gorillas provide extensive, large-scale information on the dynamics of cooperation and competition in a primate society where, as in humans, potential social partners vary greatly in the genes, developmental stage, and biological sex they share with each other. We find a selective sibling bias for prosocial behaviors, in that siblings who share matrilineal kinship affiliate at greater rates than either paternal siblings or non-siblings, and that this bias weakens as individuals become more distant in age. While such a result is consistent with a wide range of previous research, none of the reasons proposed for this selective bias in primates appear to apply to our population: mountain gorillas gain regular exposure to siblings of all types, across life stages; furthermore, patterns of aggressive behavior, in contrast to affiliation, suggest that mountain gorillas can in fact recognize paternal siblings, though they evidently do not favor them as cooperative partners. Ultimately, our study underscores a diversity of means, some evidently yet to be revealed, through which individuals might perceive and engage in sibling relationships to achieve fitness outcomes.

## Materials and methods

Our study subjects came from a population of habituated wild mountain gorillas living in Volcanoes National Park, Rwanda, that have been monitored nearly continuously for the last 54 years by the Dian Fossey Gorilla Fund. Using focal follow and scan data collected by researchers and staff, we compiled a dataset of all available dyadic gorilla behavior spanning the years of 2003–2017. We

then supplemented this dataset with demographic and relatedness data (for maternal relatedness, via direct observation; for paternal relatedness, via genetic paternity determination–see e.g. *Vigilant et al., 2015*) on individuals pulled from long-term records. From this combined dataset, we excluded interactions with infants <1 year of age at time of observation, parent-offspring interactions, and interactions between dyads for which we could not calculate relatedness from available data. This yielded a final, curated dataset containing 157 unique individuals studied from late infancy to adulthood (75 F, 82 M; average age at time of observation = 9.75 years; age range: 1–38.5 years old) and 38,996 total hours of observation.

## Composition of dyads

Our dataset of behavior from 157 individuals contained 1934 unique dyad pairs. Of these dyads, 1235 shared neither a mother nor father ('non-siblings'), 555 shared a father but not a mother ('paternal siblings'), and 43 shared both a mother and a father ('full siblings'). In addition to dyads known to share a mother but not a father (n=50), there were a number of dyads with the same mother, but with paternity data missing for one or both individuals (n=51). To maximize sample size, we combined these two groups into the category of 'maternal siblings'; due to this analytic choice, this category can be effectively conceived of as 'at least maternal siblings'. See *Appendix 2—Tables 1 and 2* for analyses using only confirmed maternal siblings, which were very similar to those reported below. Mixed-sex dyads were the most common sex category in our dataset (n=977), followed by male-male (n=503) and female-female (n=454). Dyads differed in age by an average of 5.85 years (SD: 4.53 years; range: 0–23.5 years); for reference, the average interbirth interval in mountain gorillas is 3.9 years (*Eckardt et al., 2016*). We used this continuous age difference variable as our primary index of familiarity between individuals, following a number of previous studies on primate kinship (e.g., *Widdig et al., 2001*; *Pfefferle et al., 2014*; *Wikberg et al., 2014*). While we had information on shared group membership in early life, which could also serve as a potential index of familiarity within dyads, we do not focus on this variable in our primary analyses, as it did not allow us to disambiguate between relatedness and familiarity–dyads of individuals who grew up in different natal groups were virtually never (n=3) siblings in our dataset. However, as a robustness check of our main findings, we also performed supplementary analyses on the subset of dyads who lived in the same social group during the first year of the younger partner's life; i.e., those who had substantial early-life familiarity with one another (n=6,724 dyad-years). We report the correspondence between these analyses and our primary models in our main results section, and we provide full output of these secondary models in our appendix (see *Appendix 1—Tables 1 and 2*).

## Behavioral measures

We evaluated five different categories of dyadic behaviors as outcome variables: grooming, playing, non-contact aggression, contact aggression, and time spent in close (2 m) proximity. We operationalized these behaviors from standardized definitions used in previous publications about this gorilla population (see e.g. *Rosenbaum et al., 2015*). Trained observers regularly undergo interobserver reliability tests. The former four behavioral categories were evaluated as counts (corrected for exposure time; see Data analysis) within the dyad during focal observations, regardless of directionality, while the latter category of time in close proximity was evaluated by counting the number of instantaneous scan samples in which a dyad was observed within 2 m of each other (also corrected for exposure time). Across primates, a substantial body of work has investigated how rates of these social behaviors might change with age (e.g. play: *Fagen, 1993*; aggression: *Del Giudice et al., 2009*; *Kulik et al., 2015*; *Grebe et al., 2019*; grooming: *Almeling et al., 2016*; *Schino and Pinzaglia, 2018*). General age-related trends in our dataset corroborate previously established patterns such as large play decreases, large aggression increases, and moderate grooming decreases as average age within the dyad increases (*Appendix 3—figure 1*). Importantly, our primary results pertaining to relatedness, sex configuration, and age differences are robust to the inclusion of age as a covariate; see *Appendix 3—Tables 1 and 2*.

## Data analysis

We conducted all analyses in R (version 4.1.2). Our main statistical models for each behavioral outcome consisted of cross-classified generalized linear mixed models (conducted using the *glmmTMB* package;

*Brooks et al., 2017*) these models included separate random intercept terms for each individual within the dyad and the dyad itself, in addition to random slope terms for relatedness, age difference, and sex makeup within dyads. Given low incidences of many behaviors, we aggregated behaviors into annual counts, making the dyad-year the fundamental unit of analysis (total n=7832). Even with annual aggregation, instances of aggression were uncommon. Therefore, counts of contact and non-contact aggression were summed into a single category for analysis (see *Appendix 4—figures 1 and 2*; *Appendix 4—Tables 1 and 2* for results with individual aggression categories, which were qualitatively similar to those reported below).

In models predicting each behavioral outcome, we included terms for relatedness, age difference, and sex makeup, as well as two-way interactions between relatedness and sex makeup, relatedness and age difference, and sex makeup and age difference. As mothers plausibly mediate many of the social behaviors we examined, especially early in life, we also included the proportion of observations with mothers in close proximity (i.e. the average proportion between the two members of the dyad for that observation year), and this variable's interaction with relatedness, as covariates in all models. We did not include the natal group structure of the dyad (i.e. single versus multi-male) due to the small number of dyads we had available in which both individuals grew up single-male groups (n=10 dyads, 60 dyad-years), and due to the lack of variability of relatedness–all dyads in these groups were paternal siblings.

In models containing significant main effect or interaction terms, we decompose omnibus comparisons and report targeted marginal effects and contrasts using the *emmeans* package (*Lenth, 2022*), with all reported p-values corrected for false discovery rate. We modeled our count outcomes as rates with a negative binomial family in *glmmTMB* and an offset term for exposure time (either logged hours of observation, or logged sum of scans for both individuals, per dyad-year). Regression coefficients (γ) are reported as changes in the log of the outcome variable with each unit increase in the predictor variable. For each model, we verified model fit by inspecting the deviation, dispersion, and outliers of residuals using the *DHARMa* package (*Hartig, 2022*). All data and code necessary to reproduce our results are available publicly at https://osf.io/6qgj5.

## Acknowledgements

The authors are grateful to Winnie Eckardt and Robin Morrison for their expertise and assistance with data. We thank the Rwandan government and the Rwanda Development Board for their long-term support of the research, monitoring, and protection activities of The Dian Fossey Gorilla Fund's Karisoke Research Center. We are deeply indebted to all Karisoke field staff for their tireless support in collecting long-term behavioral and demographic data. This study was supported by the University of Michigan, the NSF Graduate Research Fellowship Program and Doctoral Dissertation Improvement Grant No 1122321, and the donors who support The Dian Fossey Gorilla Fund.

## Additional information

### Funding

| Funder | Grant reference number | Author |
| --- | --- | --- |
| National Science Foundation | 1122321 | Stacy Rosenbaum |
| Dian Fossey Gorilla Fund | | Nicholas M Grebe<br>Jean Paul Hirwa<br>Tara S Stoinski<br>Linda Vigilant<br>Stacy Rosenbaum |

The funders had no role in study design, data collection and interpretation, or the decision to submit the work for publication.

## Author contributions
Nicholas M Grebe, Conceptualization, Data curation, Formal analysis, Investigation, Visualization, Methodology, Writing - original draft, Writing - review and editing; Jean Paul Hirwa, Conceptualization, Data curation, Formal analysis, Investigation, Writing - review and editing; Tara S Stoinski, Resources, Supervision, Funding acquisition, Project administration, Writing - review and editing; Linda Vigilant, Funding acquisition, Investigation, Methodology, Writing - review and editing; Stacy Rosenbaum, Conceptualization, Data curation, Formal analysis, Supervision, Funding acquisition, Investigation, Visualization, Methodology, Writing - original draft, Writing - review and editing

## Author ORCIDs
Nicholas M Grebe http://orcid.org/0000-0003-1411-065X
Stacy Rosenbaum http://orcid.org/0000-0002-7576-5302

## Decision letter and Author response
Decision letter https://doi.org/10.7554/eLife.80820.sa1
Author response https://doi.org/10.7554/eLife.80820.sa2

## Additional files

### Supplementary files
• MDAR checklist

### Data availability
All data and code necessary to reproduce our results are available publicly at https://doi.org/10.17605/OSF.IO/6QGJ5.

The following dataset was generated:

| Author(s) | Year | Dataset title | Dataset URL | Database and Identifier |
|---|---|---|---|---|
| Grebe NM, Hirwa JP, Stoinski TS, Vigilant L, Rosenbaum S | 2022 | Dynamics of cooperation and competition in mountain gorilla siblings | https://doi.org/10.17605/OSF.IO/6QGJ5 | Open Science Framework, 10.17605/OSF.IO/6QGJ5 |

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

# Appendix 1

Results for analyses restricted to dyads with "early-life familiarity" (see Materials and methods for details of categorization; total dyad-years: 6724). Full model results available from data and code posted publicly at https://osf.io/6qgj5.

**Appendix 1—table 1.** Omnibus statistics for target parameters. Effects p<0.05 bolded.

| | Effect (early life familiarity only) | | | | | |
| | Relatedness | Sex category | Age differences | Rel×sex category | Rel×age diff | Sex ×age diff |
|---|---|---|---|---|---|---|
| *Play* | $F(3, 6682) =$ 1.61, p = 0.184 | **$F(2, 6682) =$ 5.79, p = 0.003** | **$F(1, 6682) =$ 167.31, p < 0.001** | $F(6, 6682) =$ 1.06, p = 0.387 | $F(3, 6682) =$ 0.10, p = 0.963 | **$F(2, 6682) =$ 3.47, p = 0.031** |
| Grooming | **$F(3, 6682) =$ 24.15, p < 0.001** | **$F(2, 6682) =$ 3.83, p = 0.022** | **$F(1, 6682) =$ 5.28, p = 0.022** | $F(6, 6682) =$ 1.12, p = 0.347 | $F(3, 6682) =$ 0.94, p = 0.419 | **$F(2, 6682) =$ 9.56, p < 0.001** |
| Proximity | **$F(3, 6682) =$ 43.90, p < 0.001** | **$F(2, 6682) =$ 3.58, p = 0.028** | **$F(1, 6682) =$ 23.33, p < 0.001** | $F(6, 6682) =$ 0.36, p = 0.904 | **$F(3, 6682) =$ 11.56, p < 0.001** | $F(2, 6682) =$ 0.26, p = 0.771 |
| Aggression | $F(3, 6682) =$ 0.18, p = 0.910 | $F(2, 6682) =$ 0.67, p = 0.510 | **$F(1, 6682) =$ 34.49, p < 0.001** | $F(6, 6682) =$ 0.97, p = 0.444 | $F(3, 6682) =$ 0.59, p = 0.621 | **$F(2, 6682) =$ 4.46, p = 0.012** |

**Appendix 1—table 2.** Estimated marginal means and standard errors across relatedness and sex categories.

| | Relatedness | | | | Sex Category | | |
| | Full siblings | Maternal half | Paternal half | Non-siblings | Female-female | Male-male | Mixed-sex |
|---|---|---|---|---|---|---|---|
| Play | 2.35 (0.40) | 1.72 (0.33) | 0.99 (0.09) | 1.33 (0.10) | 1.16 (0.16) | 2.23 (0.26) | 1.36 (0.15) |
| Grooming | 1.94 (0.43) | 1.44 (0.27) | 0.22 (0.03) | 0.27 (0.03) | 1.57 (0.19) | 0.30 (0.05) | 0.56 (0.07) |
| Proximity | 42.5 (3.6) | 41.4 (2.9) | 14.2 (0.7) | 15.5 (0.6) | 30.4 (1.7) | 22.8 (1.4) | 22.4 (1.2) |
| Aggression | 0.41 (0.07) | 0.44 (0.08) | 0.55 (0.03) | 0.59 (0.03) | 0.39 (0.05) | 0.62 (0.06) | 0.48 (0.04) |

# Appendix 2

Results for analyses using a 'stricter' categorization of maternal siblings (n = 50; see Materials and methods for details of categorization)

**Appendix 2—table 1.** Omnibus statistics for target parameters.
Effects p<0.05 bolded.

| | Effect (strict relatedness categories) | | | | | |
| | Relatedness | Sex category | Age differences | Rel×sex category | Rel×age diff | Sex ×age diff |
|---|---|---|---|---|---|---|
| Play | $F(3, 7586)$ = 1.23, p = 0.296 | $F(2, 7596)$ = 2.65,p = 0.071 | $F(1, 7586)$ = 121.42, **p < 0.001** | $F(6, 7596)$ = 2.60, **p = 0.016** | $F(3, 7586)$ = 1.01, p = 0.387 | $F(2, 7596)$ = 5.66, **p = 0.003** |
| Grooming | $F(3, 7586)$ = **15.28, p < 0.001** | $F(2, 7596)$ = 2.82, p = 0.060 | $F(1, 7586)$ = 3.74, p = 0.053 | $F(6, 7596)$ = 1.36, p = 0.229 | $F(3, 7586)$ = **2.73, p = 0.043** | $F(2, 7596)$ = **8.26, p < 0.001** |
| Proximity | $F(3, 7586)$ = **31.67, p < 0.001** | $F(2, 7596)$ = 2.34, p = 0.097 | $F(1, 7596)$ = **16.75, p < 0.001** | $F(6, 7596)$ = 1.30, p = 0.255 | $F(3, 7586)$ = **18.83, p < 0.001** | $F(2, 7596)$ = 0.25, p = 0.778 |
| Aggression | $F(3, 7586)$ = 0.14, p = 0.936 | $F(2, 7596)$ = 0.19, p = 0.825 | $F(1, 7596)$ = **21.96, p < 0.001** | $F(6, 7596)$ = **2.17, p = 0.043** | $F(3, 7586)$ = 0.16, p = 0.927 | $F(2, 7596)$ = **5.21, p = 0.006** |

**Appendix 2—table 2.** Estimated marginal means and standard errors across relatedness and sex categories.

| | Relatedness | | | | Sex category | | |
| | Full siblings | Maternal half | Paternal half | Non-siblings | Female-female | Male-male | Mixed-sex |
|---|---|---|---|---|---|---|---|
| Play | 2.35 (0.43) | 2.16 (0.43) | 1.08 (0.11) | 1.04 (0.08) | 1.23 (0.17) | 2.45 (0.31) | 1.22 (0.16) |
| Grooming | 2.19 (0.46) | 1.67 (0.41) | 0.26 (0.03) | 0.31 (0.03) | 1.55 (0.19) | 0.39 (0.08) | 0.68 (0.09) |
| Proximity | 43.4 (3.7) | 42.1 (4.0) | 15.1 (0.7) | 17.7 (0.7) | 31.4 (1.8) | 23.5 (1.6) | 25.0 (1.5) |
| Aggression | 0.47 (0.08) | 0.62 (0.14) | 0.61 (0.04) | 0.75 (0.04) | 0.53 (0.07) | 0.70 (0.09) | 0.60 (0.06) |

# Appendix 3

Results for analyses including average age as a covariate in analyses.

**Appendix 3—table 1.** Omnibus statistics for target parameters.
Effects p<0.05 bolded.

| | Effect (average age adjusted) | | | | | | |
| --- | --- | --- | --- | --- | --- | --- | --- |
| | Relatedness | Sex category | Age differences | Average age | Rel.×sex category | Rel.×age diff. | Sex ×age diff. |
| Play | $F(3, 7782) = 5.92$, p < 0.001 | $F(2, 7782) = 6.86$, p = 0.001 | $(1, 7782) = 105.95$, p < 0.001 | $F(1, 7782) = 859.52$, p < 0.001 | $F(6, 7782) = 1.22$, p = 0.294 | $F(3, 7782) = 1.94$, p = 0.121 | $F(2, 7782) = 8.99$, p < 0.001 |
| Grooming | $F(3, 7782) = 23.14$, p < 0.001 | $F(2, 7782) = 5.74$, p = 0.003 | $F(1, 7782) = 3.20$, p = 0.074 | $F(1, 7782) = 11.83$, p < 0.001 | $F(6, 7782) = 2.16$, p = 0.044 | $F(3, 7782) = 3.31$, p = 0.019 | $F(2, 7782) = 6.68$, p = 0.001 |
| Proximity | $F(3, 7782) = 39.59$, p < 0.001 | $F(2, 7782) = 3.79$, p = 0.023 | $F(1, 7782) = 31.58$, p < 0.001 | $F(1, 7782) = 23.46$, p < 0.001 | $F(6, 7782) = 1.14$, p = 0.339 | $F(3, 7782) = 25.41$, p < 0.001 | $F(2, 7782) = 0.10$, p = 0.907 |
| Aggression | $F(3, 7782) = 0.24$, p = 0.871 | $F(2, 7782) = 0.35$, p = 0.703 | $F(1, 7782) = 40.38$, p < 0.001 | $F(1, 7782) = 103.42$, p < 0.001 | $F(6, 7782) = 1.55$, p = 0.158 | $F(3, 7782) = 0.11$, p = 0.957 | $F(2, 7782) = 4.73$, p = 0.009 |

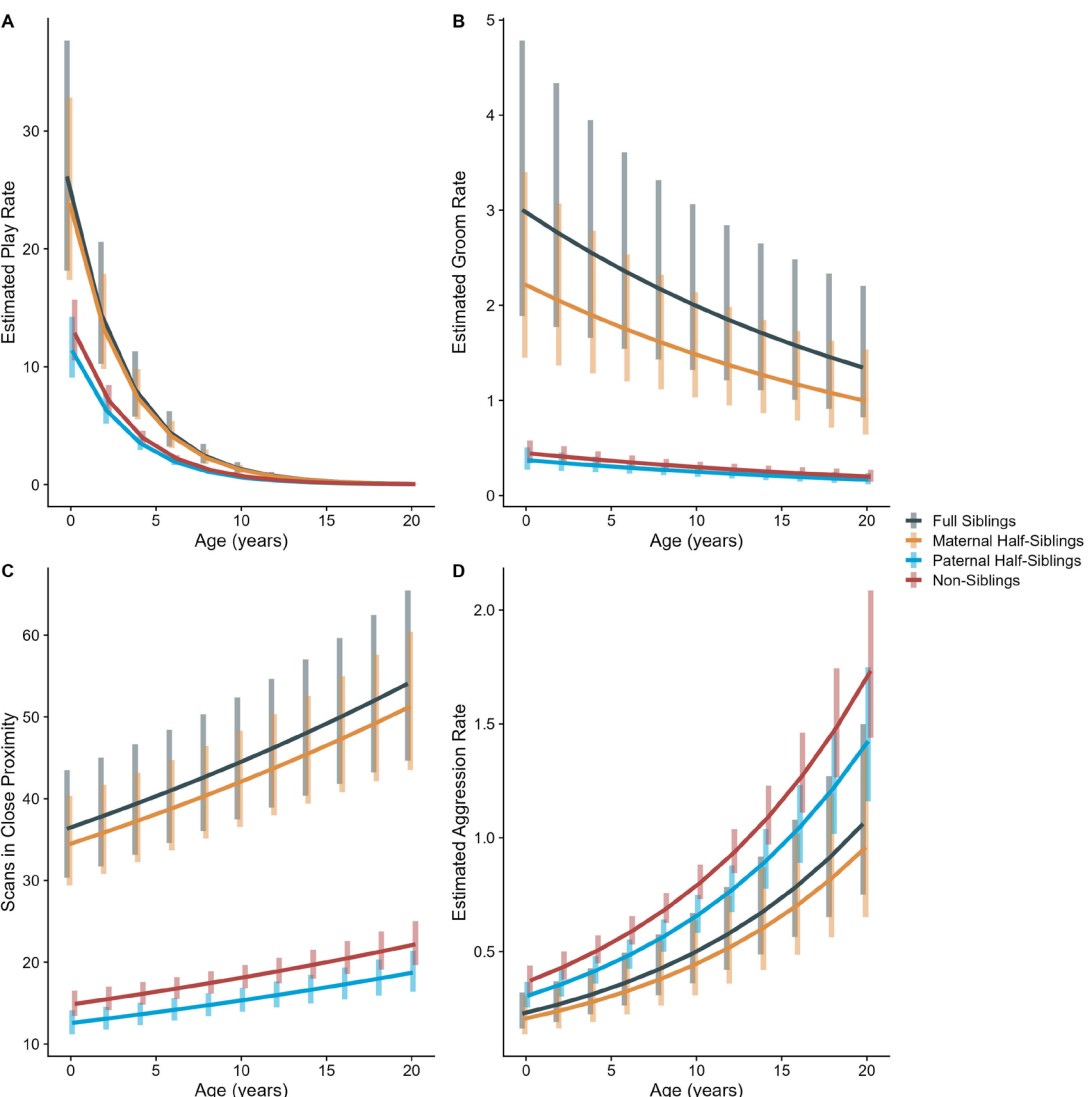

**Appendix 3—figure 1.** Estimated trends of (**A**) play, (**B**) grooming, (**C**) close proximity, and (**D**) aggression across average age of gorilla dyads. Trends adjusted for all parameters listed in **Appendix 3—table 1**. Bars represent 95% CI for rates of behavior at a given age .

**Appendix 3—table 2.** Estimated marginal means and standard errors across relatedness and sex categories, adjusting for average age.

|  | Relatedness | | | | Sex category | | |
|---|---|---|---|---|---|---|---|
|  | Full siblings | Maternal half | Paternal half | Non-siblings | Female-female | Male-male | Mixed-sex |
| Play | 1.77 (0.29) | 1.61 (0.22) | 0.77 (0.06) | 0.87 (0.05) | 0.74 (0.08) | 2.20 (0.21) | 1.00 (0.10 ) |
| Grooming | 2.08 (0.45) | 1.54 (0.29) | 0.26 (0.03) | 0.31 (0.03) | 1.61 (0.19) | 0.36 (0.06) | 0.62 (0.08) |
| Proximity | 44.6 (4.3) | 41.3 (2.9) | 15.1 (0.7) | 17.9 (0.7) | 31.6 (1.7) | 24.1 (1.5) | 24.2 (1.4) |
| Aggression | 0.46 (0.07) | 0.42 (0.08) | 0.62 (0.04) | 0.75 (0.04) | 0.45 (0.05) | 0.66 (0.07) | 0.55 (0.05) |

# Appendix 4

Results for analyses assessing contact aggression and non-contact aggression separately.

**Appendix 4—table 1.** Omnibus statistics for target parameters. Effects p<0.05 bolded.

| | Effect (separate aggression categories) | | | | | |
| | Relatedness | Sex category | Age differences | Rel.×sex category | Rel.×age diff. | Sex ×age diff. |
|---|---|---|---|---|---|---|
| Contact aggression | $F(3, 7783) =$ 2.72, **p = 0.042** | $F(2, 7596) =$ 1.02, p = 0.359 | $F(1, 7596) =$ 14.29, **p < 0.001** | $F(6, 7596) =$ 2.11, **p = 0.049** | $F(3, 7586) =$ 0.72, p = 0.541 | $F(2, 7596) =$ 2.19, p = 0.113 |
| Non-contact aggression | $F(3, 7783) =$ 0.28, p = 0.834 | $F(2, 7596) =$ 2.71, p = 0.067 | $F(1, 7596) =$ 7.11, **p = 0.008** | $F(6, 7596) =$ 2.09, p = 0.052 | $F(3, 7586) =$ 0.11, p = 0.955 | $F(2, 7596) =$ 1.33, p = 0.264 |

**Appendix 4—table 2.** Estimated marginal means and standard errors across relatedness and sex categories.

| | Relatedness | | | | Sex Category | | |
| | Full siblings | Maternal half | Paternal half | Non-siblings | Female - female | Male - male | Mixed - sex |
|---|---|---|---|---|---|---|---|
| Contact aggression | 0.25 (0.04) | 0.19 (0.05) | 0.31 (0.02) | 0.35 (0.02) | 0.22 (0.03) | 0.29 (0.04) | 0.30 (0.03) |
| Non-contact aggression | 0.12 (0.04) | 0.18 (0.05) | 0.18 (0.02) | 0.23 (0.02) | 0.21 (0.03) | 0.18 (0.03) | 0.14 (0.02) |

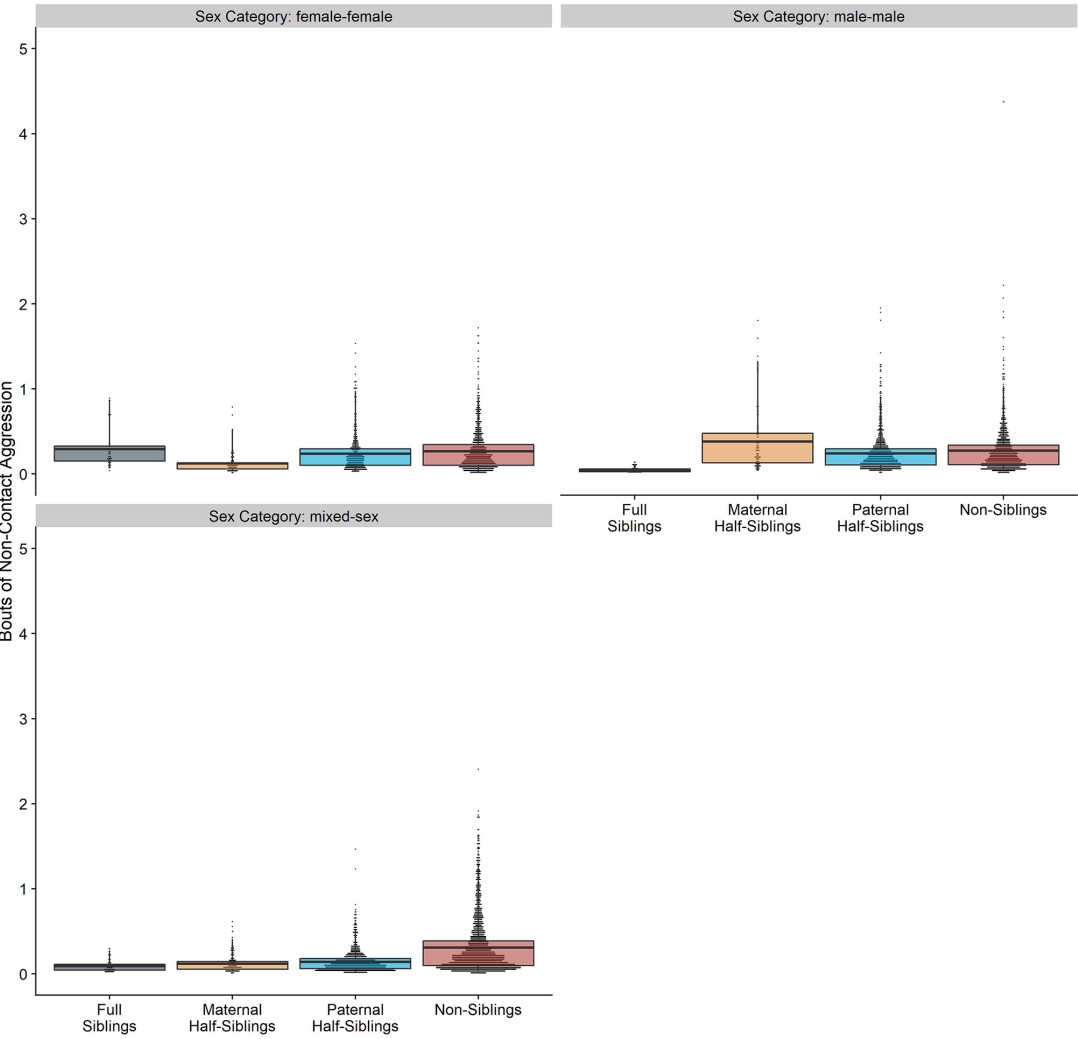

**Appendix 4—figure 1.** Box and dot plots showing estimated non-contact aggression within gorilla dyads, separated by relatedness and sex category.

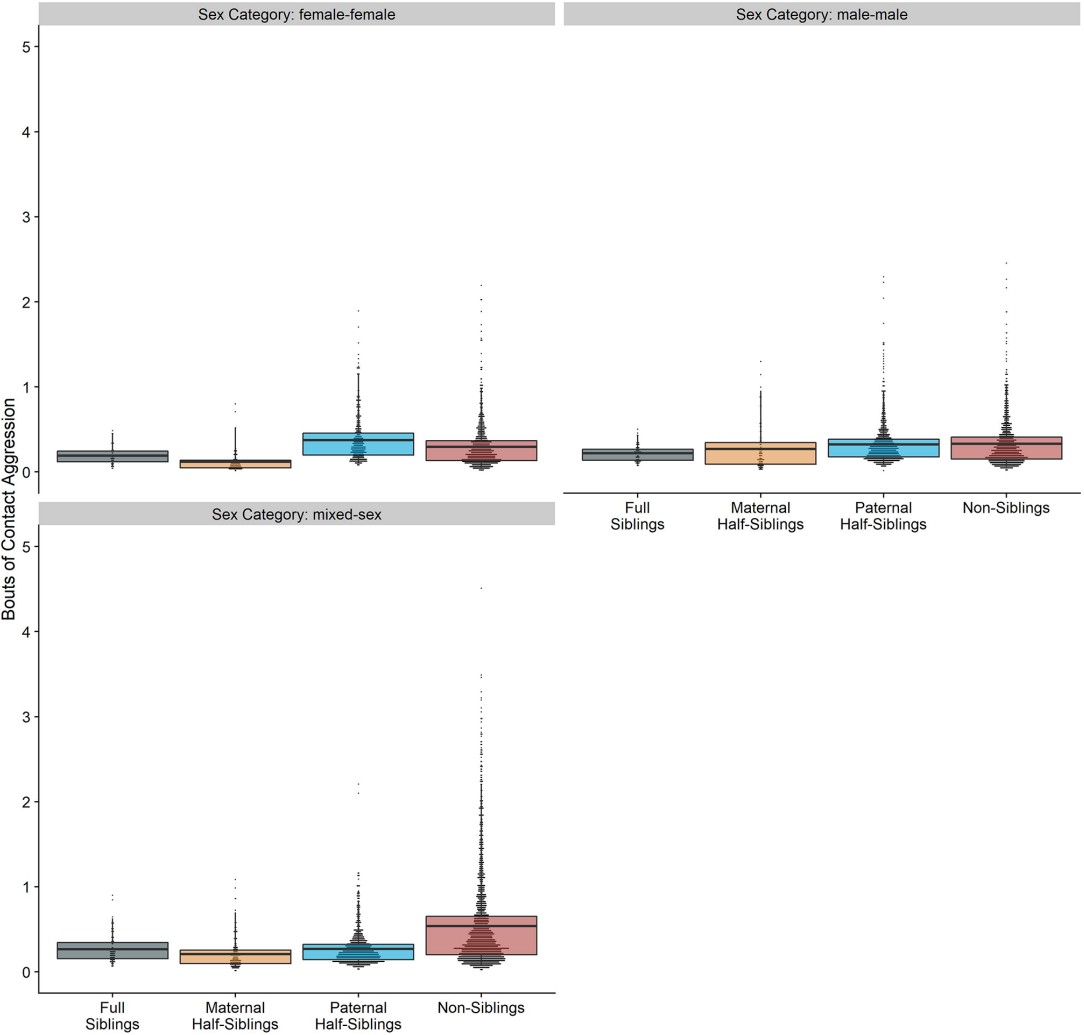

**Appendix 4—figure 2.** Box and dot plots showing estimated contact aggression within gorilla dyads, separated by relatedness, and sex category.

