## [Editor Report]

This study fundamentally advances our understanding of the ways in which kin recognition might operate in complex social societies using a compelling longitudinal dataset of a wild mountain gorilla population. Relying on detailed behavioural and pedigree data for 157 individuals, and robust statistical analyses, the authors clearly show affiliative biases among maternal siblings with aggression being more likely for unrelated siblings. Given the structure of mountain gorilla society, this research calls into question the assumptions underlying maternal kin preferences, particularly when paternal kin may be distinguishable, and therefore provides a valuable contribution to discussions on mechanisms of kin selection and kin discrimination.

---

## [Decision Letter]

**Decision letter after peer review:**

Thank you for submitting your article "Neither kin selection nor familiarity explain affiliative biases towards maternal siblings in wild mountain gorillas" for consideration by *eLife*. Your article has been reviewed by 3 peer reviewers, and the evaluation has been overseen by a Reviewing Editor and George Perry as the Senior Editor. The reviewers have opted to remain anonymous.

Essential revisions:

All three reviewers and I read this manuscript with great interest. This study has the potential to provide a strong and novel contribution to our understanding of kin selection in the primate lineage but only if the following major concerns can be addressed.

1. The operationalization of familiarity as 'age difference' is problematic for a number of reasons. I would suggest the authors avoid using the term familiarity (in the title and elsewhere) and instead simply use the term 'age difference' when appropriate. However, given the importance of familiarity with this topic, it will still need to be introduced and discussed as an alternative explanation for the results (see all three Reviewer comments for details).

2. Can the authors include random slopes in their analyses? Such a large dataset would be ideal, if not necessary, to fit random slopes in their models and it could have a critical influence on the results.

3. Given that there are both single-male and multi-male groups in this analysis could you please (a) test/control for the influence of natal group structure (single or multi-male) in your analysis and (b) add the natal group structure to your csv file for each individual gorilla? We think this is critical to the study, given that in single male groups (especially those with long silverback tenure), virtually everyone would be paternally related therefore the space for kin selection to operate would be significantly reduced (see also Review #2 for detailed comments).

4. The authors need to address how the social behaviours they examine vary with the age of individuals and how this might affect their results (see also Reviewers #1 and #3 for detailed comments).

5. The authors need to provide clear predictions as to why kin selection would even benefit mountain gorilla siblings, and for both sexes. How are mountain gorilla competition and cooperation possibly affected? Please go beyond the general theory and provide specific expectations based on the social system of these mountain gorillas (see also Reviewers #2 and #3 for detailed comments). Also please summarize the main results of critical studies cited in the introduction to ease readership for a non-specialist audience.

*Reviewer #1 (Recommendations for the authors):*

I think that this paper represents an impressive use of the long-term data at Karisoke. My main concern with the main arguments in the paper is that the discussion leaves little room for a simpler familiarity mechanism and, as the analysis stands, the argument that gorillas can recognize their paternal kin, specifically, has not been thoroughly tested. I understand the constraint that there were very few out-group related pairs, and that dominant males do sire a majority of offspring, but what about in-group unrelated dyads? If the sample is large enough, then this work stands the best chance of putting a finger on the mechanism. If not, that's fine too, but the limitation should be expanded upon in the discussion.

A few other analytical points: the first is one that I mention in the public review about age. I apologize that I don't have a specific idea of the absolute best way to deal with the problem. I would be satisfied with any evidence you can provide in the main body or the supplementary that age-related changes in social behavior weren't a major shaping force on affiliation/aggression between dyads.

Social behaviors were aggregated into annual counts – was there any variation in social behaviors between years either overall or within-dyads? My assumption would be that if there were lean years (which I know is relaxed for gorillas) or a group takeover or break-up that this might have a large effect on rates of affiliation and aggression in a given year. Usually, the year might be included as a main effect or a random term to account for any variation that was due to events like that, so I was hoping the authors could clarify here to explain the choice to exclude such a term.

L373-376: Could you please clarify the term for mother's proximity here? Is it the proportion of scans that the mother was in proximity to both partners in a dyad, either partner in a dyad, etc.?

L34-38: These last few sentences are a little clunky and I had to reread them a few times to sort out what the authors meant. Please consider rewording to make the points more clear.

*Reviewer #2 (Recommendations for the authors):*

My main recommendations would be to:

– Make clearer predictions about expected patterns of affiliation and aggression, e.g. replacing the questions in the paragraph on lines 144-152 with predictions based on the functions of the different behaviors and how they relate to kinship (e.g. aggression in mixed-sex dyads should be lower among kin because it reflects mating). i think this would be useful to go beyond a simplistic notion of 'higher r = more affiliation, less aggression', which then leads to the conclusion that 'kin selection does not explain affiliation'.

– Reconsider the operationalization of familiarity, especially for maternal kin. I just find it hard to believe statements like "sibling and non-sibling age-mates in the same social group would typically be expected to possess close familiarity" (lines 236-237). Surely, the mother is a main focal point for all young gorillas, and they would get more exposure to their maternal siblings than to others simply by virtue of sharing the mother's proximity? As mentioned before, the age difference measure is also somewhat problematic because there cannot be an age difference of 0 (or even <2-3y) for maternal sibs (I assume there weren't many twins), but there can be for paternal sibs; in fact, that's when paternal sibship is most likely.

*Reviewer #3 (Recommendations for the authors):*

In a revised version I suggest the authors provide more detail about the nature and function of social interactions (including cooperative ones) in mountain gorillas and how they change during development.

In explanations for why maternal but not paternal kin bias occurs in mountain gorillas, one important measure to consider is that a mother may be more permissive of one of her own elder offspring vs. other similarly-aged group members, interacting with her dependent offspring (which might influence bonding between gorillas and their younger group mates – even when mothers are not in close proximity).

This is not a hypothesis that requires testing here, but I think it could be helpful to speak to this – if relevant to the behavior of the study species – in the Discussion. (How often are female gorillas aggressive when others play roughly with their infants or juveniles, for example? If their adult daughter vs. another adult female carried their infant away to play with them would the mother's behavior be different?)

Last, I understand that the title is correct with respect to core kin selection theory, but I personally found it slightly misleading, because the results do find a kin bias, simply a maternal-kin bias and familiarity (though clearly defined as a predictor within the paper) has a vague broader meaning for the title. It refers particularly to familiarity within maternal siblings of varying age (age difference), but it is easily interpreted to mean familiarity on a broad level (e.g. any individuals in the same natal group – which could not be included in models because it was too correlated with other variables). As a suggestion, it might be clearer to make the title something like "Mountain gorillas demonstrate affiliative biases toward maternal but not paternal siblings of all ages."

[Editors’ note: further revisions were suggested prior to acceptance, as described below.]

Thank you for resubmitting your work entitled "Mountain gorillas maintain strong maternal affiliative biases despite high male reproductive skew and extensive exposure to paternal kin" for further consideration by *eLife*. Your revised article has been evaluated by George Perry (Senior Editor) and a Reviewing Editor.

The manuscript has been much improved but there are some remaining issues that need to be addressed, primarily concerning edits to the text to aid clarity of certain terms, the sampling, and results.

Please take into account the detailed recommendations outlined below by two reviewers when you revise your article:

*Reviewer #2 (Recommendations for the authors):*

I appreciate all the changes the authors made to the manuscript and find it much improved. The analyses and results are compelling as they are, and my comments are merely intended to further clarify the "packaging". Sorry that the comments turned out to be rather long, it's really not a huge issue but just took some time to explain.

Main comment: From kin selection to kin recognition to expressed kin preferences – I can't help but feel like the argument sometimes comes across as a bit backward, e.g. in the abstract where a preference for maternal kin (over paternal kin) occurs 'despite high relatedness certainty', whereas normally low skew underlies such 'biases' (lines 34-38). This sounds like gorillas lack something, or got something "wrong", though really it's the case that most other primates lack something (high skew) and therefore can't possibly recognize paternal kin, and are limited to maternal kin preferences – hence the bias is borne out of necessity, not because it is somehow preferable. To me at least, the flow of the argument generally goes more like this: From a theoretical perspective, we would predict individuals to preferentially cooperate with kin due to inclusive fitness benefits, which should select for kin recognition whenever possible, though in practice maternal kin recognition is much easier than paternal kin recognition (because one can use simple cues like associating with the same adult female, while paternity uncertainty in polygamous mating systems obviates the use of simple cues). Paternal kin recognition only becomes feasible (in the absence of some kind of phenotypic matching) when one can use similarly simple cues like associating with the same adult male, which becomes more reliable the fewer adult males there are and the higher the reproductive skew. So gorillas have all the ingredients for paternal kin recognition (at least historically), which sets up the expectation in the introduction of paternal kin preferences (over non-kin). yet the results show that they don't seem to express these preferences (with the discussion focusing on why they don't express these preferences, e.g. mismatches in group structure derailing paternal kinship cues, or paternal kin not typically offering cooperative benefits). I know the authors are aware of all this (and in some places the argument is very clear), and I'm sorry for rambling, but I just feel like the argument could be laid out a bit more clearly in some places. Take it or leave it!

*Reviewer #3 (Recommendations for the authors):*

This revision is much improved from the previous version. I appreciated all the changes the authors have made in responses to all editor and reviewer comments.

I still have several concerns regarding age changes/developmental perspectives in this study:

First I think the manuscript is strengthened by the supplemental analyses that incorporate the age of individuals in each dyad for each year.

However the authors still need to explicitly inform readers that their study includes both immature and mature individuals and provide context for how siblings relationships may change during age that are rooted in gorilla socioecology.

I appreciate that this is understudied:

Here is an additional resource that provides descriptive details on how social interactions between gorillas and their mothers compared to other group members manifest during different developmental stages.

Watts, David P., and Anne E. Pusey. "Behavior of juvenile and adolescent great apes." Juvenile primates: Life history, development, and behavior (2002): 148-167

I also realize that documenting behavioral shifts in development are not the focus of this study.

But not providing basic information about age ranges and what we know and do not know about how relationships change with age limits and even misguides how readers unfamiliar with the study system will interpret findings. For example they may even assume you studied only adult sibling pairs as is true in many of the key kin selection studies cited in the introduction.

In addition, it is not only that behaviors change with age (which the authors address well in this revision) but that dynamics of dyadic relationships do.

For example, as authors touch upon in their introduction, in baboons and chimpanzees sister-brother sibling relationships can be critical to survival and/or improve fitness while one sibling is immature, but not in adulthood based on dispersal and/or species-typical dominance/cooperation patterns (e.g. Engh et al., 2009; Hobaiter et al., 2014).

---

## [Author Response]

Essential Revisions (for the authors):All three reviewers and I read this manuscript with great interest. This study has the potential to provide a strong and novel contribution to our understanding of kin selection in the primate lineage but only if the following major concerns can be addressed.1. The operationalization of familiarity as 'age difference' is problematic for a number of reasons. I would suggest the authors avoid using the term familiarity (in the title and elsewhere) and instead simply use the term 'age difference' when appropriate. However, given the importance of familiarity with this topic, it will still need to be introduced and discussed as an alternative explanation for the results (see all three Reviewer comments for details).

In response to this comment, and related ones from reviewers, we’ve made a number of changes. We summarize these changes here, but we also respond in detail to specific reviewer comments below.

First, we’ve changed the title to “Mountain gorillas maintain strong maternal affiliative biases despite high male reproductive skew and extensive exposure to paternal kin”, to avoid mentioning familiarity before readers have a chance to understand what we mean by the term.

Next, we recognize that age difference is an imperfect measure, but we feel that it is the best choice amongst the available options (the other being shared natal group). We now explain more about why we think age difference is a useful index of familiarity. First, there is precedent for its use in the literature; second, it is useful for disambiguating between shared natal group and relatedness; and third, mountain gorillas of similar ages spend considerable time in age-structured ‘play groups’ when they are young:

Lines 439-443: “We used this continuous age difference variable as our primary index of familiarity between individuals, following a number of previous studies on primate kinship (e.g., Widdig et al., 2001; Pfefferle et al., 2014; Wikberg et al., 2014). While we had information on shared group membership in early life, which could also serve as a potential index of familiarity within dyads, we do not focus on this variable in our primary analyses, as it did not allow us to disambiguate between relatedness and familiarity–dyads of individuals who grew up in different natal groups were virtually never (n = 3) siblings in our dataset.”

Lines 295-299: “Due to this highly cohesive structure, all individuals in a group, related or not, are very likely quite familiar with one another. Even within this tight-knit context, however, ubiquitous 'play groups' of infants and juveniles frequently form around adult males (Rosenbaum and Silk, 2022), providing similar-aged animals even more opportunities to interact and become familiar with one another.”

Third, we report robustness checks for our main analyses in which we only examine dyads who had “early life familiarity” (i.e., its members belonged to the same group when the younger member was born, and the dyad members were still in the same group at the younger member’s first birthday), in response to Reviewer 1’s comments (see our Response #6):

Lines 443-448: “However, as a robustness check of our main findings, we also performed supplementary analyses on the subset of dyads who lived in the same social group during the first year of the younger partner’s life; i.e. those who had substantial early-life familiarity with one another (n = 6724 dyad-years). We report the correspondence between these analyses and our primary models in our main Results section, and we provide full output of these secondary models in our supplementary materials (see Tables S1 – S2).”

Results for affiliation remain quite similar, while results for aggression change appreciably. We’re grateful for the reviewer’s suggestion for this additional analysis, as we feel it provides valuable context to our results–namely, that even when individuals have extensive exposure to each other, relatedness matters for patterns of affiliation. When it comes to aggression, on the other hand, much of it seems to be driven by males aggressing against unrelated females who transfer into a group (i.e., females who are both unfamiliar and unrelated). We now provide this additional context in two locations (lines 272-275, 351-356):

“Inspection of our data confirmed that much aggression specifically occurred in the context of females transferring into groups–that is, when females first encountered unrelated, unfamiliar males–accounting for non-significant pairwise differences among dyads with early-life familiarity.”

And

“Our interpretation is further bolstered by the distinct statistical patterns that emerge for aggression when examining unrelated mixed-sex dyads with early-life familiarity, which suggest much of the mixed-sex aggression in our data took place among unrelated and unfamiliar male-female dyads. These encounters were most common when females transferred into new groups, and were thus likely to be particularly attractive mates for males.”

2. Can the authors include random slopes in their analyses? Such a large dataset would be ideal, if not necessary, to fit random slopes in their models and it could have a critical influence on the results.

Thank you for raising this point. Indeed, we had originally included maximal random-slopes terms in our models, and though we opted to abandon them in our original submission because of consistent non-convergence issues, this comment led us to revisit the issue. In our revised analyses, we’ve followed an approach that balances the arguments made by Barr et al., (2013) and Matuschek et al., (2017). We begin with the ‘maximal’ random effects model for dyads, and as necessary, we remove individual terms to achieve convergence. The upshot is that all of our models now include random slopes and intercepts, with the correlation between these terms omitted to permit convergence. The qualitative pattern of results from these models is nearly identical to before, with some minor differences in pairwise comparisons. These changes are tracked and reported in the results, and the methods section was updated to clarify that we include both random slopes and intercepts.

Barr, D. J., Levy, R., Scheepers, C., and Tily, H. J. (2013). Random effects structure for confirmatory hypothesis testing: Keep it maximal. *Journal of Memory and Language*, *68*(3), 255-278.

Matuschek, H., Kliegl, R., Vasishth, S., Baayen, H., and Bates, D. (2017). Balancing Type I error and power in linear mixed models. *Journal of Memory and Language, 94*, 305-315.

3. Given that there are both single-male and multi-male groups in this analysis could you please (a) test/control for the influence of natal group structure (single or multi-male) in your analysis and (b) add the natal group structure to your csv file for each individual gorilla? We think this is critical to the study, given that in single male groups (especially those with long silverback tenure), virtually everyone would be paternally related therefore the space for kin selection to operate would be significantly reduced (see also Review #2 for detailed comments).

Thanks for this comment. Indeed, as we state in the introduction, mountain gorillas regularly form single-male and multi-male social groups. Interestingly, for a combination of potential reasons—including (a) chance variation during our study period, and (b) the much smaller number of offspring that result from single-male groups in general—there is a comparatively tiny number of rows in our dataset where one or both members of a dyad grew up in a single-male group (210 and 60 rows, respectively, out of 7832; this variable is now included in the public dataset as requested). Additionally, for this n=60 subset (10 dyads in total, because most dyads are represented across multiple years) where both partners had a single-male natal group, all of the dyads were paternal siblings, meaning we have no relatedness variation. There were zero full sibling, maternal sibling, or unrelated dyads. Given these constraints of the data, we do not feel that we can make any meaningful claims about the effect of natal group structure on sibling relationships. Karisoke currently follows several single-male groups, so we hope that this is something we will be able to follow up on in the future when more data from these groups becomes available.

We have now included a paragraph about this limitation in the discussion (lines 333-345):

“One important limitation of our analyses is that we have far less information about sibling relationships in single-male groups—where shared paternity is nearly or entirely certain—than we do about siblings in multi-male groups, where reproductive skew is high but not 100%. The structure of the research groups during our 14 years of data was primarily multi-male, with only 60 dyad-years of data (n=10 total dyads) from single-male groups. All of these dyads were paternally related (i.e., there were no unrelated or maternally related dyads available in single-male groups). This small sample size and lack of variability in relatedness precludes a test of whether natal group structure predicts features of sibling relationships. If all available partners are paternal kin, then discrimination becomes a question of full versus paternal siblings, rather than discrimination among four categories (full, half maternal, half paternal, and unrelated). Choice is constrained, perhaps limiting the utility of a discrimination mechanism. It would be interesting to know whether animals in single-male groups treat paternal versus full siblings differently, but the currently available data do not allow us to answer this question.”

Additionally, we added a sentence to the methods section to clarify why this variable was not included in the models (lines 483-486):

“We did not include the natal group structure of the dyad (i.e., single versus multi-male) due to the small number of dyads we had available in which both individuals grew up single-male groups (n=10 dyads, 60 dyad-years), and due to the lack of variability of relatedness–all dyads in these groups were paternal siblings.”

4. The authors need to address how the social behaviours they examine vary with the age of individuals and how this might affect their results (see also Reviewers #1 and #3 for detailed comments).

Please see our Response #7, where we address this point in detail. Briefly, while the rates of these behaviors do indeed change as animals age, accounting for this effect does not meaningfully change the primary results we report. We now include supplementary analyses demonstrating this, and we reference them in our methods section.

5. The authors need to provide clear predictions as to why kin selection would even benefit mountain gorilla siblings, and for both sexes. How are mountain gorilla competition and cooperation possibly affected? Please go beyond the general theory and provide specific expectations based on the social system of these mountain gorillas (see also Reviewers #2 and #3 for detailed comments). Also please summarize the main results of critical studies cited in the introduction to ease readership for a non-specialist audience.

We’ve extensively addressed both the issue of predictions, and the issue of summarizing studies, in our response to individual reviewers (see e.g. our Response #11). To summarize our changes to the first issue, we’ve added two paragraphs to the introduction that clarifies the potential dynamics of kin selection opportunities in gorillas specifically (lines 140-162):

“This means that siblings can be an important source of support well into adulthood. For example, adult males benefit greatly from allies who help them fend off outsider challengers and prevent females from transferring (Sicotte, 1993, Rosenbaum et al., 2016, Mirville et al., 2018). Adult females with offspring benefit from male protection from infanticide (Harcourt and Stewart, 2007, Robbins et al., 2013); in multi-male groups, this protection could potentially come not only from mates, but from brothers as well, who receive indirect fitness benefits from their sister’s reproductive success. While the benefits of female-female relationships remain understudied in this species, higher-ranking females have better energy balance and shorter inter-birth intervals, and it is plausible that support from other females plays a role in achieving and maintaining dominance (Robbins et al., 2005, Wright et al., 2014, Wright et al., 2020).

As noted above, however, sibling relationships are also characterized by the potential for competition as well as cooperation, and mountain gorillas are no exception. Even after co-resident mountain gorilla siblings cease competing over parental resources, they are likely to compete over other limited resources, including dominance positions, mating opportunities, and preferential foods (Harcourt and Stewart, 2007). In some other primate species, rates of aggression are as high or higher among kin than non-kin, in part because kin simply spend more time together, highlighting the complexity of making straightforward predictions about affiliative and aggressive interactions and relatedness (e.g. Silk et al., 2010). On balance, however, we expect that mountain gorillas should be more cooperative/affiliative with siblings than non-siblings due to the substantial potential for inclusive fitness benefits.”

In response to the second issue, we now elaborate on the key results of studies cited in the introduction (see in particular p.4 of our revised manuscript).

Reviewer #1 (Recommendations for the authors):I think that this paper represents an impressive use of the long-term data at Karisoke. My main concern with the main arguments in the paper is that the discussion leaves little room for a simpler familiarity mechanism and, as the analysis stands, the argument that gorillas can recognize their paternal kin, specifically, has not been thoroughly tested. I understand the constraint that there were very few out-group related pairs, and that dominant males do sire a majority of offspring, but what about in-group unrelated dyads? If the sample is large enough, then this work stands the best chance of putting a finger on the mechanism. If not, that's fine too, but the limitation should be expanded upon in the discussion.

Thanks for this comment. The issue of untangling familiarity and relatedness is one that our team spent considerable time discussing when writing the original draft. Familiarity has been assessed in a number of different ways in previous studies, including, as the reviewer suggests, by shared group membership in early life. When deciding on analyses, we did create a binary variable that assesses “early familiarity”—a dyad was coded as “yes” if its members belonged to the same group when the younger member was born, and if the members were still in the same group at the younger member’s first birthday. The vast majority of rows in the dataset (84%) consist of dyads with early familiarity. Ultimately, because of the confound between this variable and relatedness (which the reviewer notes), we decided to instead focus on age differences as our primary measure. Age differences are also frequently used to assess familiarity in other studies of primate kinship (see e.g. Widdig et al., 2002; Pfefferle et al., 2014; Wikberg et al., 2014; we now cite these studies where we further explain our use of age differences as an index of familiarity in the introduction).

Regardless, we are happy to oblige the reviewer’s request and only examine dyads who had early-life familiarity. When re-running our analyses on just these dyads, results are qualitatively very similar for affiliation: full and maternal siblings play, groom, and spend more time together than paternal siblings or non-siblings. Interestingly, patterns of aggression are qualitatively different: mixed-sex non-siblings no longer stand out as sources of high aggression. This suggests that much of this aggression occurs within “unfamiliar” mixed-sex dyads, i.e. when females transfer into new groups. We’re grateful for the reviewer suggesting this analysis, as we think it provides some very interesting context for interpreting our results. Thus, we report these new analyses in two locations of the revised manuscript: first, as a supplementary robustness check for our results; and second, as context for our observed patterns of aggressive behavior (e.g. lines 351-356).

A few other analytical points: the first is one that I mention in the public review about age. I apologize that I don't have a specific idea of the absolute best way to deal with the problem. I would be satisfied with any evidence you can provide in the main body or the supplementary that age-related changes in social behavior weren't a major shaping force on affiliation/aggression between dyads.

We’re happy to have the opportunity to address this point. Indeed, it is well-established from previous studies that the rates of social behaviors change with age. Play generally becomes much less frequent, aggression generally becomes more frequent, and patterns of grooming appear to reflect interactions between age and other factors. We find similar patterns in our dataset: average age within the dyad has a massive negative effect on play rates; a marginally significant negative effect on grooming; a positive effect on the frequency of close proximity to others; and a very large positive effect on aggression. Given that these results for age do not meaningfully deviate from previously established patterns, and that they were not the focus of our paper, we elected not to discuss them in the main text. But, to address the reviewer’s concern that these patterns might significantly alter our primary results–something readers might also wonder–we now report the results of models including average age in our supplementary materials (Tables S5 and S6, Figure S4). The upshot is that our results stay consistent, even when controlling for these age-related changes that are sometimes quite large. In other words, the differences we describe in our results are over and above general age trends, which we now note in our Methods section (lines 458-465):

“Across primates, a substantial body of work has investigated how rates of these social behaviors might change with age (e.g. play: Fagen, 1993; aggression: Del Giudice et al., 2009; Kulik et al., 2015; Grebe et al., 2019; grooming: Almeling et al., 2016; Schino and Pinzaglia, 2018). General age-related trends in our dataset corroborate previously established patterns such as large play decreases, large aggression increases, and moderate grooming decreases as average age within the dyad increases (Table S5, Figure S4). Importantly, our primary results pertaining to relatedness, sex configuration, and age *differences* are robust to the inclusion of age as a covariate; see Tables S5 – S6.”

Social behaviors were aggregated into annual counts – was there any variation in social behaviors between years either overall or within-dyads? My assumption would be that if there were lean years (which I know is relaxed for gorillas) or a group takeover or break-up that this might have a large effect on rates of affiliation and aggression in a given year. Usually, the year might be included as a main effect or a random term to account for any variation that was due to events like that, so I was hoping the authors could clarify here to explain the choice to exclude such a term.

We originally chose not to include a random effects term for year for a few reasons: first, because gorillas don’t share the same environmental constraints that lead to the ‘lean years’ that are, for example, seen during droughts with Amboseli baboons; second, while it’s certainly plausible that social upheaval could affect rates of social behavior, our impression from years in the field with these groups is that behavioral disruption is fairly transient; and finally, we believe year-to-year variation is already captured to a large extent by the repeated sampling of individuals and dyads across multiple years. Any residual variance accounted for by year would have to be ‘on top of’ individual- or dyad-specific variance, and we just didn’t expect there to be very much of that. That said, the reviewer may still wonder about the effect of including year as a random effect. We re-ran our primary analyses including this term and found it did not affect any substantive conclusions–these results are available as part of our code posted publicly on OSF.

L373-376: Could you please clarify the term for mother's proximity here? Is it the proportion of scans that the mother was in proximity to both partners in a dyad, either partner in a dyad, etc.?

Certainly. We now clarify that the “mom’s proximity” variable is coded as the proportion of scans in that year in which an individual’s mother was in close proximity, averaged between the two individuals in the dyad.

L34-38: These last few sentences are a little clunky and I had to reread them a few times to sort out what the authors meant. Please consider rewording to make the points more clear.

We’ve now reworded these lines in our abstract.

Reviewer #2 (Recommendations for the authors):My main recommendations would be to:– Make clearer predictions about expected patterns of affiliation and aggression, e.g. replacing the questions in the paragraph on lines 144-152 with predictions based on the functions of the different behaviors and how they relate to kinship (e.g. aggression in mixed-sex dyads should be lower among kin because it reflects mating). i think this would be useful to go beyond a simplistic notion of 'higher r = more affiliation, less aggression', which then leads to the conclusion that 'kin selection does not explain affiliation'.

During the writing of the introduction we debated about the extent to which we should advance specific predictions. For the most part, we did not feel that we could do so in a principled manner; given the well-documented complexity of sibling relationships in a variety of species, it is easy to find support for virtually any prediction one cares to make (as just one example: it would be quite easy to justify predicting that male-male, male-female, or female-female sibling dyads should be the closest type of sibling dyads, or alternatively, that any of those should engage in the most aggression; it simply depends on what assumptions one makes about the value of what they can offer one another relative to the strength of the competition they impose). Therefore, we elected to refrain from making specific predictions and instead (1) examine the interactions between our predictor variables of interest, and (2) phrase our inquiry as questions to be answered, rather than predictions to be supported/refuted. Please note that we also added some explanatory language to the introduction that we hope helps to clarify why we used this approach, as well as to address reviewer requests for additional details about the specifics of gorilla dynamics, specifically (lines 151-162):

“As noted above, however, sibling relationships are also characterized by the potential for competition as well as cooperation, and mountain gorillas are no exception. Even after co-resident mountain gorilla siblings cease competing over parental resources, they are likely to compete over other limited resources, including dominance positions, mating opportunities, and preferential foods (Harcourt and Stewart, 2007). In some other primate species, rates of aggression are as high or higher among kin than non-kin, in part because kin simply spend more time together, highlighting the complexity of making straightforward predictions about affiliative and aggressive interactions and relatedness (e.g. Silk et al., 2010). On balance, however, we expect that mountain gorillas should be more cooperative/affiliative with siblings than non-siblings due to the substantial potential for inclusive fitness benefits.”

As a final note, there was disagreement among the reviewers about the approach we took to this problem. Reviewer 3 notes that “A second major strength is the opportunity this dataset and study system provides to test predictions about proposed mechanisms for kin recognition in primates. The authors do a good job of making these details about their study system and their predictions clear.” Therefore, we hope that in the revision we have managed to strike a balance between not making predictions we do not feel are necessarily theoretically justified, while making it clear what questions we are proposing to answer and why.

– Reconsider the operationalization of familiarity, especially for maternal kin. I just find it hard to believe statements like “sibling and non-sibling age-mates in the same social group would typically be expected to possess close familiarity” (lines 236-237). Surely, the mother is a main focal point for all young gorillas, and they w”uld get more exposure to their maternal siblings than to others simply by virtue of sharing the mother’s proximity?

We’re happy to address this comment. Mothers undoubtedly mediate social interactions to some extent, which was our rationale for including the intensity of her presence, and its interaction with the relatedness of the target dyad, as covariates. In the public review, the reviewer expresses skepticism that this “fully accounts for the ways mothers structure one’s social network, increasing exposure among maternal sibs throughout one’s lifetime”. We agree that there may well be forms of early exposure that we’re unable to assess, such as the sharing of night nests or preferential mother-father relationships—both of which we noted on p. 18 of the original manuscript.

At the same time, we stand by our argument that mountain gorilla groups are remarkably close-knit, which should lead to close familiarity among siblings and non-siblings alike. It has been observed for decades that “[Mountain] gorilla groups are quite cohesive in that the members rarely drift far from each other. The diameter of feeding or resting groups is usually 200 feet or less” (Schaller and Emlen, 1963; p. 372). Often, the focal point for social interactions is an adult male (whether the father or not), rather than mothers (Stewart, 2001; Rosenbaum and Silk, 2022). While it is reasonable to believe that maternal siblings may receive somewhat more exposure to one another via common proximity to their mother, even when we attempt to adjust for that variable, we (a) acknowledge this possibility and (b) have good reason to believe that mountain gorillas routinely spend large amounts of time very close to, and interacting with, both related and unrelated age-mates—especially compared to other great apes like chimpanzees.

That said, we take the point that we should qualify the claim the reviewer challenges. We have now edited this paragraph in the discussion to explain that “First, gorilla social groups are tight-knit and cohesive compared to their close ape relatives (Goodall, 1986; Remis, 1997; Morrison et al., 2021b; Schaller and Emlen, 1963). Even within this tight-knit context, however, ubiquitous ‘play groups’ of infants and juveniles frequently form around adult males (Rosenbaum and Silk, 2022), providing similar-aged animals even more opportunities to interact and become familiar with one another.” We also have added to our discussion of sharing night nests to specify that this is a manner of maternally mediated exposure we could not measure.

Stewart, KJ (2001). Social relationships of immature gorillas and silverbacks. In Robbins MM, Sicotte P, Stewart KJ, editors*. Mountain gorillas: three decades of research at Karisoke*. Cambridge, UK: Cambridge University Press, pp. 184-213.

Schaller, GB, and Emlen, JT. (1963). Observations on the ecology and social behavior of the mountain gorilla. In *African ecology and human evolution,* Routledge, pp. 368-384.

As mentioned before, the age difference measure is also somewhat problematic because there cannot be an age difference of 0 (or even <2-3y) for maternal sibs (I assume there weren’t many twins), but there can be for paternal sibs; in fact, that’s when paternal sibship is most likely.

Here, we take the reviewer to be arguing that, because certain age differences (namely, those less than two years) do not exist among full and maternal siblings, our age difference variable is likely to lead to biased estimates. It’s of course true that interbirth intervals preclude certain age differences in certain relatedness categories, but it does not follow that this makes the variable problematic for estimation. Our models estimate the effect of relatedness across the full range of age differences in our dataset, not just from 0 – 2 years, and we find that in several instances this wide range provides important moderating context to our pairwise differences across relatedness and/or sex categories. (In the figures, we depict estimates between 0 and 15 years age difference, because relatively few observations after 15 years leads to very wide confidence intervals.) We do appreciate the reviewer bringing this point up, however, in part because it alerted us to an oversight in our age difference figures—the trend lines should not extrapolate to “impossible” values for full and maternal siblings. This has now been fixed.

Reviewer #3 (Recommendations for the authors):In a revised version I suggest the authors provide more detail about the nature and function of social interactions (including cooperative ones) in mountain gorillas and how they change during development.

The first part of this suggestion is something we now address and that was also requested by other reviewers; please see our Responses #5 and #11, in which we paste the exact changes made. Regarding changes in social relationships in mountain gorillas over the course of development, this is a topic on which there is remarkably little information. In fact, the dearth of data on the ontogeny of social relationships was originally one of the motivations behind this analysis. There are some data on the ontogeny of relationships between adults and immatures, particularly parents and offspring (Fletcher, 2001; Rosenbaum et al., 2011; Rosenbaum et al., 2016), but there is very little outside of that specific context/dyad type. This is a hole in the literature that this paper is helping to fill.

Fletcher, A. W. (2001). Development of infant independence from the mother in wild mountain gorillas. Mountain gorillas: three decades of research at Karisoke.

Rosenbaum, S., Hirwa, J. P., Silk, J. B., and Stoinski, T. S. (2016). Relationships between adult male and maturing mountain gorillas (Gorilla beringei beringei) persist across developmental stages and social upheaval. *Ethology*, *122*(2), 134-150.

Rosenbaum, S., Silk, J. B., and Stoinski, T. S. (2011). Male–immature relationships in multi‐male groups of mountain gorillas (Gorilla beringei beringei). American Journal of Primatology, 73(4), 356-365.

In explanations for why maternal but not paternal kin bias occurs in mountain gorillas, one important measure to consider is that a mother may be more permissive of one of her own elder offspring vs. other similarly-aged group members, interacting with her dependent offspring (which might influence bonding between gorillas and their younger group mates – even when mothers are not in close proximity).This is not a hypothesis that requires testing here, but I think it could be helpful to speak to this – if relevant to the behavior of the study species – in the Discussion. (How often are female gorillas aggressive when others play roughly with their infants or juveniles, for example? If their adult daughter vs. another adult female carried their infant away to play with them would the mother’s behavior be different?)

This is an interesting point. In other primate species (e.g. baboons, chimpanzees), mothers are often quite intolerant of unrelated individuals handling, touching, or getting close to their infants. In mountain gorillas, our personal observation from years of experience in the field is that this is much less true. While we are not aware of any papers on this subject, unrelated individuals frequently interact even with young infants, and mothers usually only interfere if their baby shows signs of distress (e.g. whimpering or screaming), which is rare. In those cases, the offending individual typically stops what they are doing and moves away from the baby quickly, acquiescing to the mother’s authority. There is no real equivalent of the ‘kidnapping’ behavior that is common in baboons, and which likely motivates maternal suspicion of unrelated animals handling babies. While unfortunately we do not have empirical evidence to support this (yet–it would certainly be interesting to look at!), our sense is that mothers do not have strongly different reactions to relatives and non-relatives handling/interacting with their offspring. One of us (SR) also studies baboons, and she confirms that there is a very strong qualitative difference in the reaction of baboon mothers and mountain gorilla mothers to ‘outsiders’ interacting with infants.

Last, I understand that the title is correct with respect to core kin selection theory, but I personally found it slightly misleading, because the results do find a kin bias, simply a maternal-kin bias and familiarity (though clearly defined as a predictor within the paper) has a vague broader meaning for the title. It refers particularly to familiarity within maternal siblings of varying age (age difference), but it is easily interpreted to mean familiarity on a broad level (e.g. any individuals in the same natal group – which could not be included in models because it was too correlated with other variables). As a suggestion, it might be clearer to make the title something like “Mountain gorillas demonstrate affiliative biases toward maternal but not paternal siblings of all ages.”

In response to this comment and those of another reviewer and the editor (see our Response #1), we’ve changed the title of our manuscript to eliminate reference to familiarity. We still believe age difference to be an important measure of familiarity within mountain gorilla groups, but we don’t wish to mislead readers before they understand what we mean by the term. Our title is now “Mountain gorillas maintain strong maternal affiliative biases despite high male reproductive skew and extensive exposure to paternal kin”.

[Editors’ note: further revisions were suggested prior to acceptance, as described below.]

Reviewer #2 (Recommendations for the authors):I appreciate all the changes the authors made to the manuscript and find it much improved. The analyses and results are compelling as they are, and my comments are merely intended to further clarify the “packaging”. Sorry that the comments turned out to be rather long, it’s really not a huge issue but just took some time to explain.Main comment: From kin selection to kin recognition to expressed kin preferences – I can’t help but feel like the argument sometimes comes across as a bit backward, e.g. in the abstract where a preference for maternal kin (over paternal kin) occurs ‘despite high relatedness certainty’, whereas normally low skew underlies such ‘biases’ (lines 34-38). This sounds like gorillas lack something, or got something “wrong”, though really it’s the case that most other primates lack something (high skew) and therefore can’t possibly recognize paternal kin, and are limited to maternal kin preferences – hence the bias is borne out of necessity, not because it is somehow preferable. To me at least, the flow of the argument generally goes more like this: From a theoretical perspective, we would predict individuals to preferentially cooperate with kin due to inclusive fitness benefits, which should select for kin recognition whenever possible, though in practice maternal kin recognition is much easier than paternal kin recognition (because one can use simple cues like associating with the same adult female, while paternity uncertainty in polygamous mating systems obviates the use of simple cues). Paternal kin recognition only becomes feasible (in the absence of some kind of phenotypic matching) when one can use similarly simple cues like associating with the same adult male, which becomes more reliable the fewer adult males there are and the higher the reproductive skew. So gorillas have all the ingredients for paternal kin recognition (at least historically), which sets up the expectation in the introduction of paternal kin preferences (over non-kin). Yet the results show that they don’t seem to express these preferences (with the discussion focusing on why they don’t express these preferences, e.g. mismatches in group structure derailing paternal kinship cues, or paternal kin not typically offering cooperative benefits). I know the authors are awar’ of all this (and in some places the argument is very clear), and I'm sorry for rambling, but I just feel like the argument could be laid out a bit more clearly in some places. Take it or leave it!

We appreciate this thoughtful comment, and we agree that the difference in opinion here mostly comes down to framing. Regarding the term “bias”—we simply use it to mean “disproportionate weight in favor of something”, without intending to attach any sort of valence to it. In terms of gorillas ‘lacking’ something (which the reviewer mentions here and in a later comment), we once again did not intend this to highlight a deficiency—we were just attempting to draw a contrast between the social features of better-studied primate species, and mountain gorillas, who do not possess those same features.

However, we understand that the inadvertent side effect might have been to imply a deficiency. To remove any suggestion of this, we edited our discussion (lines 352-354) to simply state “Interestingly, this “asymmetric bias” in affiliation seems to persist even though the high reproductive skew of mountain gorillas opposes the dynamic thought to underlie such biases in other primate species.”

We elected to keep our language largely the same in the abstract (minus some small changes that we detail in responses below), for the reason we outline above: the established idea is that low reproductive skew leads to maternal kin biases only, and it is this idea that our results challenge, as we see (mostly) similar biases in a primate with high reproductive skew. The reviewer might ask, “why not flip the comparison, and say mountain gorillas are expected to have kin discrimination abilities that other primates lack?” Our response is that we don’t necessarily think that’s the case. To detail our thinking, we’ve added a paragraph to the introduction (line 180-196) explaining how one could make an argument that high skew makes paternal kin discrimination achievable, or that it makes it unnecessary. Our results are generally more consistent with the latter, though we also find some evidence for the former (regarding aggression). Going in, we weren’t confident which outcome was more likely. For us, the key takeaway is that our results in many ways resemble those from e.g. baboons, even though gorilla society is quite different from baboon society, and that’s something that calls for revisiting prominent theoretical accounts of kin selection.

Reviewer #3 (Recommendations for the authors):This revision is much improved from the previous version. I appreciated all the changes the authors have made in responses to all editor and reviewer comments.I still have several concerns regarding age changes/developmental perspectives in this study:First I think the manuscript is strengthened by the supplemental analyses that incorporate the age of individuals in each dyad for each year.However the authors still need to explicitly inform readers that their study includes both immature and mature individuals and provide context for how siblings relationships may change during age that are rooted in gorilla socioecology.I appreciate that this is understudied:Here is an additional resource that provides descriptive details on how social interactions between gorillas and their mothers compared to other group members manifest during different developmental stages.Watts, David P., and Anne E. Pusey. “Behavior of juvenile and adolescent great apes.” Juvenile primates: Life history, development, and behavior (2002): 148-167I also realize that documenting behavioral shifts in development are not the focus of this study.But not providing basic information about age ranges and what we know and do not know about how relationships change with age limits and even misguides how readers unfamiliar with the study system will interpret findings. For example they may even assume you studied only adult sibling pairs as is true in many of the key kin selection studies cited in the introduction.In addition, it is not only that behaviors change with age (which the authors address well in this revision) but that dynamics of dyadic relationships do.For example, as authors touch upon in their introduction, in baboons and chimpanzees sister-brother sibling relationships can be critical to survival and/or improve fitness while one sibling is immature, but not in adulthood based on dispersal and/or species-typical dominance/cooperation patterns (e.g. Engh et al., 2009; Hobaiter et al., 2014).

We appreciate this critique, and we certainly wish to be responsive to it.